# Cellular response upon proliferation in the presence of an active mitotic checkpoint

Andrea Corno[1], Elena Chiroli[1], Fridolin Gross[1], Claudio Vernieri[1,2], Vittoria Matafora[1], Stefano Maffini[3] , Marco Cosentino Lagomarsino[1,4], Angela Bachi[1] , Andrea Ciliberto[1,5]

Eukaryotic cells treated with microtubule-targeting agents activate the spindle assembly checkpoint to arrest in mitosis and prevent chromosome mis-segregation. A fraction of mitotically arrested cells overcomes the block and proliferates even under persistent checkpoint-activating conditions. Here, we asked what allows proliferation in such unfavourable conditions. We report that yeast cells are delayed in mitosis at each division, implying that their spindle assembly checkpoint remains responsive. The arrest causes their cell cycle to be elongated and results in a size increase. Growth saturates at mitosis and correlates with the repression of various factors involved in translation. Contrary to unperturbed cells, growth of cells with an active checkpoint requires Cdh1. This peculiar cell cycle correlates with global changes in protein expression whose signatures partly overlap with the environmental stress response. Hence, cells dividing with an active checkpoint develop recognisable specific traits that allow them to successfully complete cell division notwithstanding a constant mitotic checkpoint arrest. These properties distinguish them from unperturbed cells. Our observation may have implications for the identification of new therapeutic windows and targets in tumors.

## Introduction

Cells arrest proliferation when challenged with poisons that alter microtubule–kinetochore attachment. To avoid chromosome mis-segregation, they arrest in prometaphase by activating a surveillance mechanism, the mitotic checkpoint or spindle assembly checkpoint (SAC), which inhibits the anaphase promoting complex or cyclosome (APC/C) (1). The APC/C is a multiprotein E3 ligase that catalyzes ubiquitination of proteins, thus priming them for degradation (2). In particular, two substrates of APC/C, mitotic cyclins and securin, need to be degraded for cells to progress into anaphase (3). Inhibition of APC/C, as orchestrated by the mitotic checkpoint, prolongs the duration of M-phase by stabilizing mitotic cyclins and securin. APC/C inhibition takes place through the sequestration of Cdc20, an activator of APC/C, into the so-called mitotic checkpoint complex (MCC) (4). When the checkpoint is inactive, Cdc20 activates APC/C by direct binding, giving rise to the active APC/C$^{Cdc20}$ complex. When the checkpoint is active, APC/C$^{Cdc20}$ is inhibited by MCC binding (5).

Although the mitotic checkpoint is essential in mammalian cells, it is only transiently activated during a regular cell cycle. However, specific external stimuli can induce prolonged, potentially indefinite, SAC activation. For instance, antimitotic drugs such as taxanes and vinca alkaloids (among the most used cytotoxic agents in cancer treatment) impair the proliferation of normal and cancer cells by affecting microtubule dynamics, which finally results in SAC activation. In the long run, however, the checkpoint signal cannot sustain the arrest, and cells enter anaphase even when kinetochores and microtubules are not properly attached. This phenomenon is called "adaptation" or "slippage," to emphasize the fact that cells overcome an operational checkpoint and exit the checkpoint-induced arrest (6). Cells entering into anaphase with an active SAC have higher chances that chromosome segregation has not been executed properly and that daughter cells become aneuploid.

The molecular processes taking place during a checkpoint-induced mitotic arrest have been described in some detail (6, 7, 8). In mammalian cells, slippage requires slow degradation of mitotic cyclins, which accelerates just before exit from mitosis (7). A bi-phasic arrest is also observed in yeast, where initially mitotic cyclins are stable, but are suddenly degraded when cells enter anaphase (9). Based on models and experiments in yeast, we have proposed that transition into anaphase under checkpoint activating conditions is a stochastic process, driven by random fluctuations in APC/C$^{Cdc20}$ levels (10).

After overcoming the arrest, some cells die, whereas others continue proliferating even in the constant presence of an operational mitotic checkpoint (8). In the perspective of cancer

[1]Istituto Firc di Oncologia Molecolare, Milan, Italy   [2]Medical Oncology Department, Fondazione IRCCS Istituto Nazionale Tumori, Milan, Italy   [3]Department of Mechanistic Cell Biology, Max Planck Institute of Molecular Physiology, Dortmund, Germany   [4]Physics Department, University of Milan, Milan, Italy   [5]Istituto di Genetica Molecolare, Consiglio Nazionale delle Ricerche, Pavia, Italy

Correspondence: andrea.ciliberto@ifom.eu

treatment, these are potentially dangerous cells because they go on proliferating regardless of a "stop division" signal and do so with the risk of mis-segregating chromosomes and further increasing genetic variability. On the long term, some of these cells may select specific mutations leading to stable, acquired resistance to anti-mitotics. However, on a shorter time scale, that is, during the earliest cell cycles completed in the presence of an active SAC, cells need to exploit alternative and faster solutions to deal with the stress caused by overcoming a constant "stop division" signal. How this is achieved is not currently known and in fact we do not know whether cells share similar short-term strategies or if they display different responses. The presence of specific properties would open the clinically relevant possibility of selectively targeting cells dividing under checkpoint conditions.

Here, we analyze features of *Saccharomyces cerevisiae* cells dividing with an operational checkpoint. We find that (i) they are still responsive to the mitotic checkpoint, (ii) their cell cycle network has specific synthetic interactions, (iii) they are larger than unperturbed cells, and (iv) they undergo extensive changes in protein levels.

# Results

## Two experimental approaches for the analysis of cells proliferating with an active checkpoint

To analyze cells capable to divide under checkpoint activating conditions, we induced a checkpoint signal with two different experimental approaches (Fig 1A). In the first case, we activated the checkpoint using temperature-sensitive mutants of the essential *TUB2* gene (*tub2-401*) (11). At the semipermissive temperature, microtubules are partially depolymerized (Fig S1A) and cells grow less efficiently than unperturbed wild-type cells, yet they manage to divide more efficiently than mitotic checkpoint-deficient cells impaired in microtubule polymerization (compare *tub2-401* and *tub2-401 mad2Δ* in Fig S1B). In the second experimental setup, we inhibited the transition into anaphase without disturbing microtubule-kinetochore attachment. Instead, we overexpressed the essential component of the checkpoint, Mad2, by placing three copies of the *MAD2* gene under the inducible *GAL1* promoter (*GAL1pr*) (12, 13). When cells are grown in galactose, Mad2 overexpression ectopically activates the checkpoint by inducing the formation of the MCC, thus mimicking the effect of microtubule depolymerizing drugs but without affecting microtubule dynamics (12). Also in this case, cells growing under checkpoint conditions are able to proliferate although less efficiently than unperturbed wild-type cells (Fig S1C). We observed that a small fraction of cells overexpressing Mad2 mispositions the mitotic spindle, which remains localized in the mother cells, thus impairing proper chromosome segregation. Indeed, the rate of mis-segregation of *GAL1-MAD2* cells is higher than that of unperturbed cells but lower than the rate of cells expressing *tub2-401* at the semipermissive temperature (Fig S1D, left and central panel). In particular, if we assume that all chromosomes mis-segregate with the same probability (a gross first approximation), we can use measurements

of one labeled chromosome to estimate that ~60% of *tub2-401* grown at semipermissive temperature have at least one aneuploid chromosome (see the Material and Methods section "evaluation of mis-segregation events," and Fig S1D, right panel). Using the same argument, we can infer 30% aneuploidy in *GAL1-MAD2* cells grown in galactose (Fig S1D, right panel). Whereas growth of *tub2-401* at low temperature mimics antimitotics that act on microtubule polymerization, Mad2 overexpression mirrors approaches aimed at inhibiting APC/C directly (14).

In the remaining part of the article, we will take as bona fide features of cells replicating with an active checkpoint those properties observed in both experimental systems.

## Cells proliferating with an active checkpoint mount an efficient mitotic arrest

Cells that successfully overcome the mitotic checkpoint and proliferate may be fully competent to arrest in mitosis upon recurrent checkpoint activation. Alternatively, after overcoming the SAC, they may have become refractory to the checkpoint, similarly to what has been shown for the G1 arrest induced by α-factor (15). To test these alternative possibilities, we measured the duration of mitosis in individual cells under constant checkpoint activation. We trapped cells for several hours in microfluidic chambers and recorded single-cell dynamics with live-cell imaging. Mitotic entry and anaphase onset were identified, indirectly, by the dynamics of the mitotic cyclin Clb2 tagged with GFP. Namely, the time of Clb2 accumulation marked mitotic entry, and the time of Clb2 degradation identified entry into anaphase (10) (Fig 1B). The time between mitotic entry and anaphase was used as a proxy for the extent of mitosis. The time elapsed between two budding events gave us the total cell cycle length.

After synchronization in G1, we compared cell cycle duration of wild-type cells cycling in the absence of mitotic checkpoint activation with that of cells dividing with an active checkpoint signal (Fig 1C, unperturbed [blue] and SAC-active [green], respectively). We could follow up to three cycles (generations 1, 2, and 3—gen1, 2, and 3; see Fig S2A), and throughout our analysis, we kept track of those cells that were present at the beginning of the movie. We did not include in the comparison the first cell cycle following G1 release (we call it, gen0) because it was longer than the following cycles (dotted lines in Fig 1C, gen0 SAC-active [red], and gen0 unperturbed [cyan]). We found that cells capable to divide with an active checkpoint (i.e., gen1 SAC-active cells) were similarly frequent in both experimental settings: 66% in Mad2-overexpressing cells and 72% in *tub2-401* mutants (Fig S2B).

We observed much longer mitotic durations in SAC-active cells when compared with unperturbed cells (5.3-fold increase in *tub2-401* and 3.5-fold in *GAL1-MAD2*). Cell cycle phases other than mitosis were also extended in cells dividing with an active SAC when compared with unperturbed cells, albeit less markedly (fourfold increase in *tub2-401* and 1.4-fold in *GAL1-MAD2*—see the Material and Methods section for measuring cell cycle and mitotic duration) (Fig 1D and E). The different temperatures used in the two experimental approaches (19–20°C versus 30°C) can explain the difference in fold-changes between the two experimental systems. The difference between unperturbed and SAC-active being due to

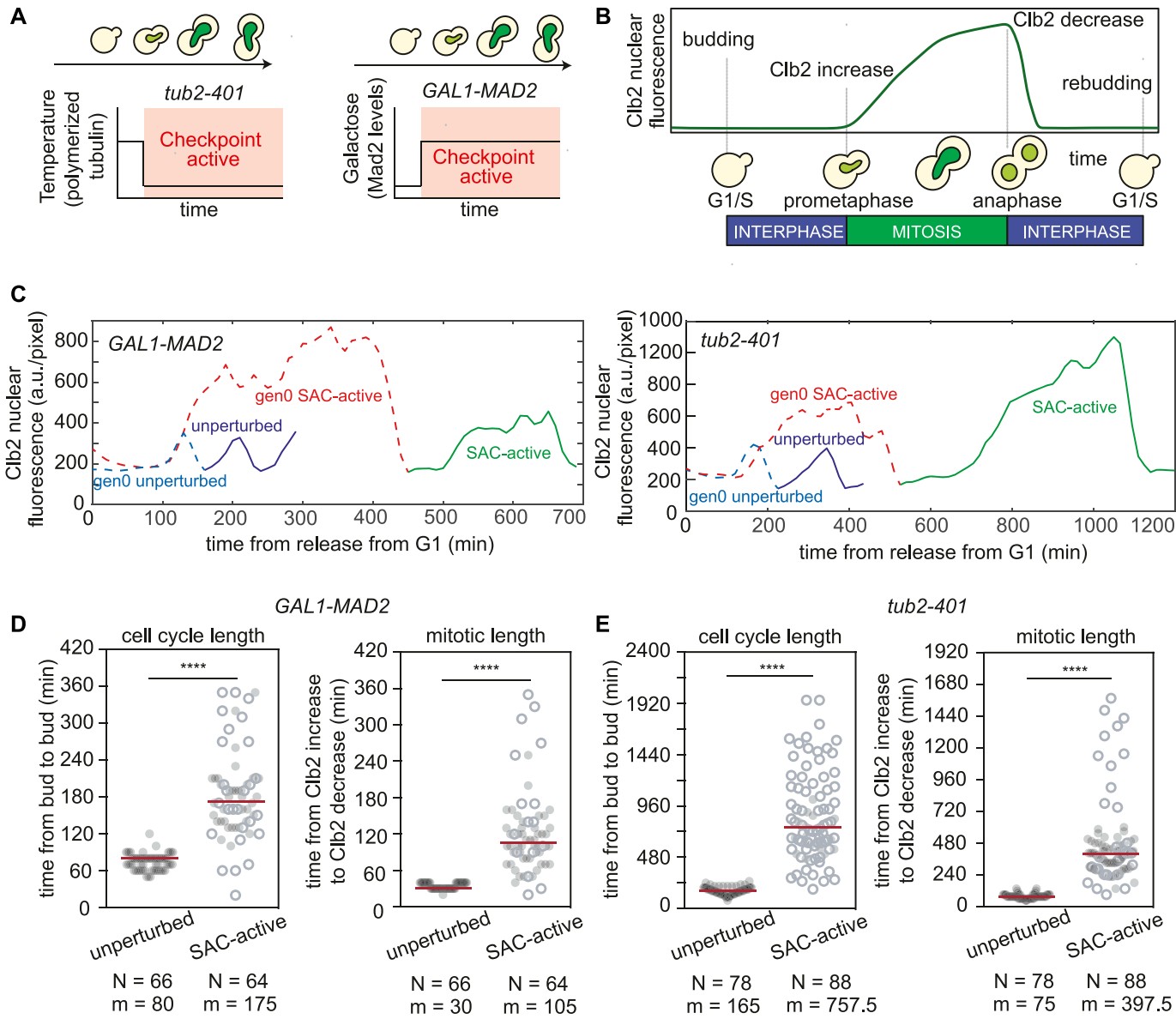

**Figure 1. Single-cell analysis of proliferating cells in the presence of an active mitotic checkpoint.**
**(A)** Schematics of the two strategies for inducing the mitotic checkpoint: perturbing the mitotic spindle (*tub2-401* grown at low temperatures, left) or inducing Mad2 overexpression (*GAL1-MAD2*, right). **(B)** Quantities measured in single-cell analysis. Budding marks the G1/S transition, increasing and decreasing levels of Clb2 correspond, respectively, to prometaphase and anaphase onset. **(C–E)** Cells were synchronized in G1 and released in the presence of galactose (yAC2006: *CLB2-GFP TUB2-mCherry*; yAC2671: *CLB2-GFP TUB2-mCherry GAL1-MAD2*) or at the semi-permissive temperature. C (yAC3491: *CLB2-GFP*, yAC2970: *CLB2-GFP, tub2-401*). **(C)** Example of Clb2-GFP trajectories for *GAL1-MAD2* cells (on the left) and for *tub2-401* cells (on the right). For Mad2-overexpressing cells, we compared cells dividing with an active checkpoint (SAC-active) to *CLB2-GFP TUB2-mCherry* released from the arrest and dividing in galactose (unperturbed). When instead the checkpoint was induced by the temperature-sensitive β-tubulin mutant, SAC-active *tub2-401 CLB2-GFP* cells were released from the G1 arrest at the semi-permissive temperature. *CLB2-GFP* grown at the semipermissive temperature were the unperturbed cells. **(D, E)** Distributions of cell cycle lengths and mitotic lengths for unperturbed and SAC-active cells under the two experimental conditions. N, number of observations; m, median of the distribution. Censored data (i.e., dead cells or precocious end of cell monitoring) were represented as white circles. Significant differences between unperturbed and SAC-active cells were evaluated with a log-rank test (significance level: 0.05).

the mitotic checkpoint is further confirmed by the fact that the cell cycle length of gen0 *tub2-401 mad2Δ* cells at the restrictive temperature was similar to that of gen0 unperturbed cells (Fig S2C). Clb2 levels increased around twofold during the prolonged arrest (Fig S2D) and went back to slightly higher levels in SAC-active cells when compared with unperturbed cells (Fig S2E). Clustering SAC-active cells by generation showed that both cell cycle and mitotic

duration remained largely unchanged as cells continued dividing under checkpoint-activating conditions (Fig S2F and G).

In summary, our results show that the duration of the cell cycle, and particularly of mitosis, is longer when cells grow with an active SAC. This suggests that cells are capable to repetitively mount a checkpoint response when grown under constant checkpoint-activating conditions.

## Synthetic interactions between constitutive checkpoint activation and *cdh1Δ*

The repetition of prolonged mitotic delays suggests that Cdc20 is inhibited during each cell cycle in SAC-active cells. Cdc20 is essential for entry into anaphase (16), but the other APC/C activator, Cdh1, can also promote cyclin B and securin degradation when Cdc20 is inhibited by the mitotic checkpoint (9). We thus asked whether Cdh1 (17)—dispensable in unperturbed cells—plays a central role in the cell cycle of SAC-active cells.

Using serial dilutions, we tested the synthetic interaction between *CDH1* deletion and constant checkpoint activation. To this purpose, *GAL1-MAD2* cells were grown in raffinose and then plated on galactose to overexpress Mad2. Notably, growth of cells overexpressing Mad2 was partially impaired when *CDH1* was deleted (Fig 2A). A similar synthetic negative effect between constant checkpoint activation and *CDH1* deletion was observed in *tub2-401* grown at restrictive temperature (Fig 2B).

It was previously reported that Cdh1 becomes essential when Clb2 levels are too high (18, 19). As such, the overexpression of Sic1—a stoichiometric inhibitor of CDK1—rescues the lack of Cdh1 in mutants that are impaired in degrading cyclin B at mitotic exit (20). Because cells dividing with an active SAC have higher average Clb2 levels when compared with unperturbed cycling cells (Fig S2D and E), we tested whether Sic1 overexpression was able to rescue the synthetic interaction observed in *cdh1Δ* mutants. Indeed, the growth of *GAL1-MAD2 cdh1Δ* cells was rescued by the presence of 10 copies of *SIC1* under the endogenous promoter (3). The result was confirmed with *tub2-401 cdh1Δ SIC1(10X)* grown at semi-restrictive temperatures, Fig 2A and B.

To confirm the role of Cdh1 in SAC-active cells, we performed single cell analysis with microfluidic devices comparing cells

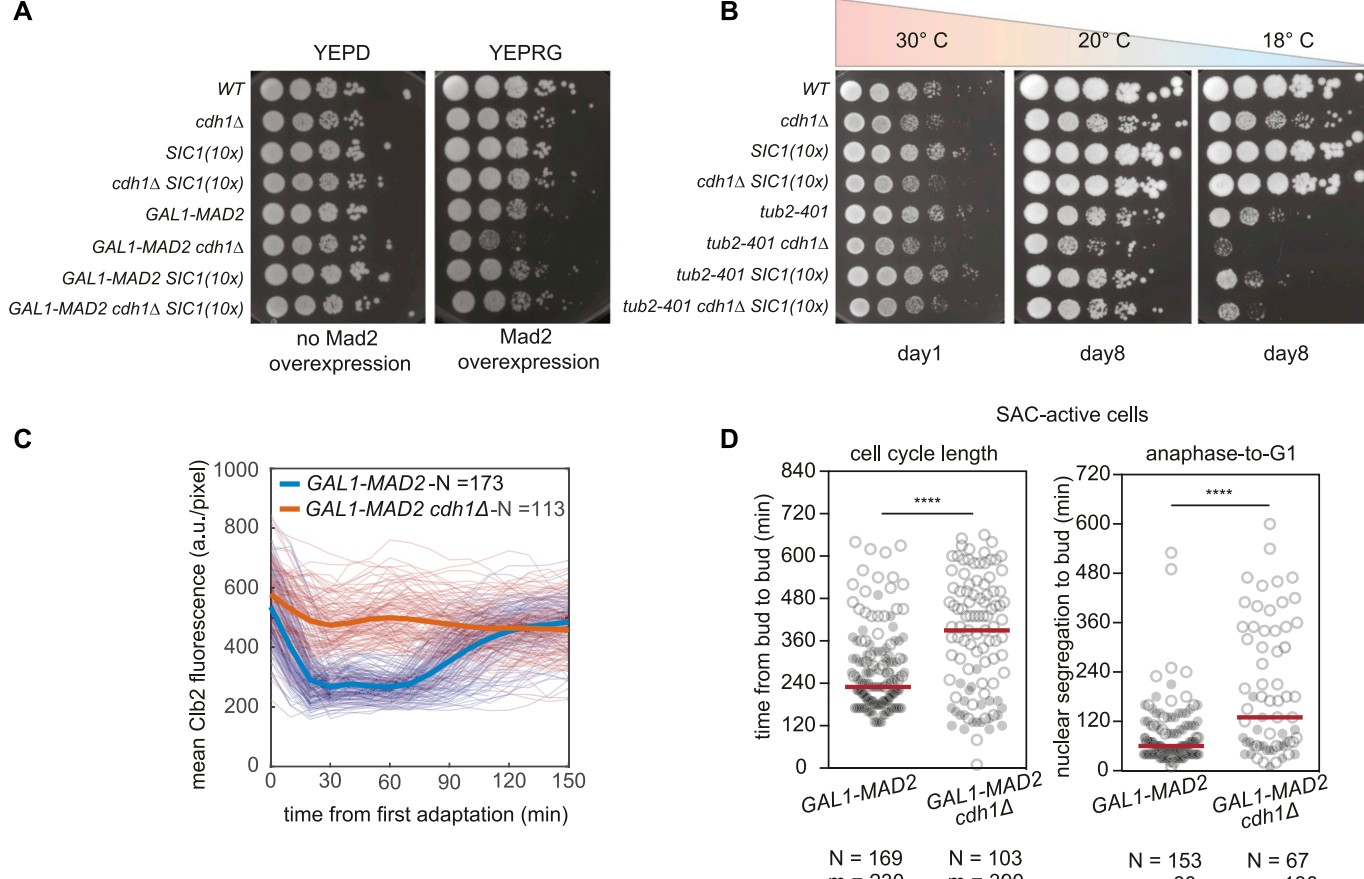

**Figure 2.   Deletion of *CDH1* impairs the proliferation of SAC-active cells.**
**(A)** Spotting of serial dilutions of *WT* (yAC3568), *cdh1Δ* (yAC1533), *SIC1(10x)* (yAC3650), *cdh1Δ SIC1(10x)* (yAC3683), *GAL1-MAD2* (yAC2465), *GAL1-MAD2 cdh1Δ* (yAC3582), *GAL1-MAD2 SIC1(10x)* (yAC3654), and *GAL1-MAD2 cdh1Δ SIC1(10x)* (yAC3659) on YEPD plate (without Mad2 overexpression) or YEPRG plate (with Mad2 overexpression). **(B)** Spotting of serial dilution of *WT* (yAC3568), *cdh1Δ* (yAC1533), *SIC1(10x)* (yAC3650), *cdh1Δ SIC1(10x)* (yAC3683), *tub2-401* (yAC3220), *tub2-401 cdh1Δ* (yAC3686), *tub2-401 SIC1(10x)* (yAC3685), and *tub2-401 cdh1Δ SIC1(10x)* (yAC3694) on YEPD plates incubated at 30°, 20°, and 18°C. **(C, D)** Single-cell analysis of *GAL1-MAD2 cdh1Δ* SAC-active cells. Proliferation of *GAL1-MAD2* (yAC3883) and *GAL1-MAD2 cdh1Δ* (yAC3885) cells carrying Htb2-mCherry and Clb2-GFP was monitored in microfluidic chamber, in the continuous presence of YEPRG medium. **(C)** Clb2 trajectories were synchronized in silico at the beginning of Clb2 degradation in the first division upon Mad2 overexpression. Individual traces of Clb2 are reported as thin lines, their mean trajectory as thick lines. N, number of trajectories. **(D)** Duration of cell cycle length and anaphase-to-G1 in SAC-active cells. Nuclear segregation was used to identify anaphase onset. N, number of observations; m, median of the distribution. Censored data (i.e., dead cells or precocious end of cell monitoring) are represented as white circles. Significant differences between *GAL1-MAD2* and *GAL1-MAD2 cdh1Δ* cells were evaluated with a log-rank test (significance level: 0.05).

overexpressing Mad2, with or without *CDH1*. Tagging Clb2 and the histone Htb2 allowed identification of the different cell cycle phases and investigation of the role of Cdh1 in Clb2 degradation.

When *CDH1* is deleted, cells cannot be synchronized in G1 (21). Hence, we had to induce a checkpoint response in asynchronous cycling cells. Upon checkpoint activation, *cdh1Δ* cells arrested in prometaphase but eventually entered anaphase with similar kinetics as *CDH1* cells (Fig S3A), confirming that overcoming the first checkpoint arrest does not require Cdh1 in budding yeast (9). However, the following cell cycles were greatly impaired in cells growing in SAC-activating conditions without Cdh1. In these cells, the fast Clb2 degradation taking place at anaphase onset observed in *CDH1* SAC-active cells (Fig 2C) did not occur, likely because APC/C$^{Cdc20}$ can only bring about a partial Clb2 degradation (16). A similar behavior was observed in *cdh1Δ* cells when the checkpoint was not activated (Fig S3B), but in this context, cells managed to proliferate efficiently (Fig 2A), likely because the levels of Clb2 are similar to that of wild types and half that of SAC-active cells (22). In the presence of high Clb2 levels, several abnormalities have been reported to occur in the next G1 phase (e.g., aberrant spindle assembly) (22). Indeed, already the first exit from mitosis (i.e., from nuclear separation to budding) took longer in *cdh1Δ* than in *CDH1* cells under SAC-activating conditions (Fig S3C). This result was confirmed in the following cycles, where *CLB2-GFP HT2B-mCherry GAL1-MAD2 cdh1Δ* cells showed prolonged anaphase-to-G1, and also longer cell cycles, Fig 2D.

In summary, we found that cells under constant checkpoint treatment display novel synthetic interactions. In particular, SAC-active cells without Cdh1 further slow down proliferation, likely because of the need to keep CDK1 activity under control.

### Cells dividing with an active SAC are larger and saturate growth for large sizes

Cell growth does not stop when the cell cycle is delayed. Accordingly, the distribution of cell sizes (i.e., of the combined cellular area of mother and daughter cells) at anaphase onset shows that SAC-active cells are on average larger than unperturbed cells, with a stable distribution of cell sizes Fig S4A. Given the large size of SAC-active cells, known active size control mechanisms (e.g., at G1/S (23)) are unlikely to play major roles. However, under the assumption of continued exponential growth, one would expect a large variability of cell sizes. We, thus, asked whether entry into anaphase in SAC-active cells was under the control of other mechanisms for size compensation.

Such mechanisms would have to compensate for the different sizes of SAC-active cells entering mitosis. In particular, cells with a smaller size at mitotic entry would have to increase their size during mitosis more than cells that entered mitosis with a larger size. In this case, we would expect a negative correlation between the relative growth of size occurring during mitosis (quantified by $G_{mitosis} = ln(A_{ANA}/A_{PM})$ in Fig 3A) and the size of cells at mitotic entry $ln(A_{PM})$ (24). This is indeed the case in both experimental systems (Fig 3B). A similar observation was made in cycling cells, in agreement with (25), although SAC-active cells show a stronger correlation (i.e., stronger size control).

Compensation could happen because cells enter anaphase when they reach a critical size or because they spend more time in mitosis when their entry size is small. In the first case, we would expect a lack of correlation between size of mitotic entry and size at anaphase; in the second case, we would expect a negative correlation between time spent in mitosis and size at entry. Neither of the two cases are fulfilled: size at anaphase is positively correlated with size at mitotic entry, (Fig S4B) and the time spent in mitosis does not correlate with size at entry (Fig S4C).

In search of alternative mechanisms to explain the observed compensation, we looked at growth throughout the cell cycle. For both SAC-active and unperturbed cells, we plotted cellular growth from budding to entry into anaphase, filtering noisy data (see the Material and Methods section and Fig S4D–F). In the analysis, we also included the first cell cycle after the release from G1 arrest (i.e., gen0 SAC-active and gen0 unperturbed in Fig 1C). For these cells, we kept track of size from α-factor release to Clb2 degradation (Fig 3A).

The area of unperturbed cells appears to increase exponentially, whereas in SAC-active cells, this regime of fast growth is followed by a slower and linear growth pattern (Fig 3C). The transition occurs primarily when Clb2 levels are high (Fig S5A), that is, before entry into anaphase, in agreement with reference (26). The growth of the whole population can be approximated by an exponential followed by a linear curve (as suggested in reference 27), where unperturbed cells grow exponentially and SAC-active cells primarily grow linearly. Individual cells released from G1 in SAC-activating conditions traverse the whole curve, starting as unperturbed and ending up similar to SAC-active cells (Fig 3C).

The interpretation of the data in Fig 3C is blurred by cell-to-cell variability. To confirm the observation independently from the absolute cell size, we plotted the local growth rate (i.e., the slope of the curve in each point, $dA/dt|_A$ (24)) as a function of cell size A, Fig S5B. We found that unperturbed cells belong to the exponential part of the curve because the local growth rate increases linearly with size ($dA/dt|_A = k*A$, in line with $A(t) \sim exp(k*t)$). By contrast, growth of SAC-active cells is saturated in the sense that the local growth rate approaches a constant value in both *tub2-401* and Mad2-overexpressing cells, regardless of the quite different shapes that these cells acquire in the two different experimental systems, Fig S5C.

These observations suggest that compensation of size may arise from a size-dependent regulation of growth (24). To support this hypothesis, we developed a simple mathematical description of the observed growth pattern consisting of an exponential and a linear regime. We assumed cell cycle lengths of unperturbed cells to be normally distributed around the measured average cycle length. Cells dividing under checkpoint-activating conditions, by contrast, were assumed to transition into anaphase in a time-independent stochastic process with an initial delay (in line with reference 10) and an overall cell cycle duration that is on average longer than for unperturbed cells (Fig 1D and E). All cells move along the same growth curve (Figs 3C and S5B) and cell division in the model simply means that the size is divided by 2 and a cell "jumps" back to the position on the growth curve that corresponds to the new size (see inset in Fig 3D, right).

In simulations (Fig 3D), cells transit between two size regimes because the cell cycle durations are picked from different

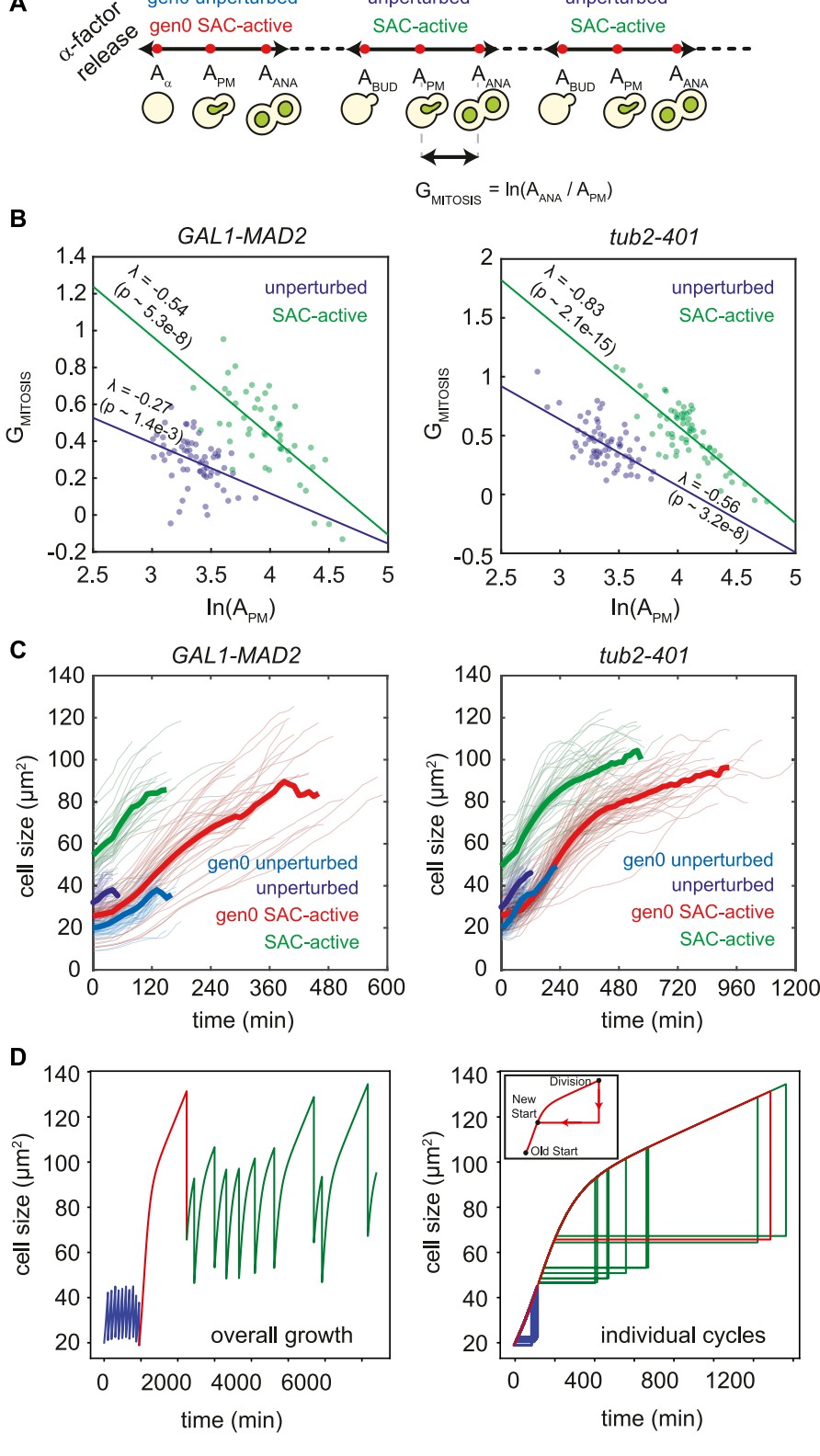

**Figure 3. SAC-active cells are larger than unperturbed cells and compensate for size.**
**(A)** Cell sizes monitored in the experiment. Size of gen0 unperturbed and gen0 SAC-active cells was monitored from the G1 release ($A_\alpha$) to Clb2 degradation (i.e., anaphase onset, $A_{ANA}$). Size of unperturbed and SAC-active cells was measured from budding ($A_{BUD}$) to $A_{ANA}$. Dashed lines indicate time intervals in which cell size was not monitored. Size when Clb2 starts to accumulate (i.e., prometaphase, $A_{PM}$) was also monitored. **(B)** Scatter plot of mitotic size multiplicative increase $G_{MITOSIS}$ (see panel A) versus cellular size at prometaphase in unperturbed and SAC-active cells (log–log). This "size-growth" plot has a slope in the presence of size control and is flat otherwise. Linear dependency was tested by performing a linear regression on data, evaluating the slope $\lambda$ (significance level: 0.05). Number of points for *GAL1-MAD2*: 65 for unperturbed, 48 for SAC-active. Number of points for *tub2-401*: 71 for unperturbed, 65 for SAC-active. **(C)** Cell size trajectories: filtered and smoothed traces are reported as thin lines and their mean as a thick line. Number of trajectories for *GAL1-MAD2*: 54 gen0 unperturbed, 65 unperturbed, 45 gen0 SAC-active, 37 SAC-active; number of trajectories for *tub2-401*: 72 gen0 unperturbed, 75 unperturbed, 78 gen0 SAC-active, 46 SAC-active. **(D)** Left: simulation of multiple cycles for an individual cell using combined exponential and linear growth. At time = 0, the distribution from which the cell cycle durations are drawn changes from unperturbed to SAC active. Right: all cycles of the simulation are arranged on one growth curve (dashed line). Inset: an example of a cell division in the plot.

distributions (a normal distribution for unperturbed cells and a delayed exponential distribution for SAC-activated cells). In particular, the cell cycle becomes longer and more variable upon SAC activation. The longer cell cycle results in a bigger size, but the relative growth rate decreases as cells become larger. After the first cycles, cells dividing with an active SAC find a relatively stable orbit

where growth and cycle times are balanced (Fig 3D) because the decrease in relative growth compensates for the variability in cycle times. Hence, in spite of a large variability in durations, the growth pattern gives rise to a stable distribution of cell sizes. Notice that the distribution of cell sizes would quickly diverge if cells were to continue exponential growth in the perturbed regime (Fig S5D).

The prediction that SAC-active cells have a larger variability in cycling times than in sizes is experimentally confirmed. The co-efficient of variation (CV) of cell cycle length in SAC-active cells is 0.39 for Mad2 overexpression and 0.47 for cells with *tub2-401*. The CVs of sizes, by contrast, are 0.19 and 0.14, respectively. The two distributions are much more similar in unperturbed cells, where the CVs for cycle time and size are 0.17 and 0.17 for *GAL1-MAD2*, and 0.21 and 0.17 in *tub2-401*, respectively.

In summary, our data show that SAC-active cells are larger than unperturbed cells and saturate their growth in mitosis; this peculiar growth rate may act as an effective size control mechanism in SAC-active cells. This is recapitulated by a simple model where (i) overcoming the SAC is a random event; (ii) cells grow exponentially when small and slow down to linear growth when they become larger; and (iii) cell cycle length is on average longer in SAC-induced cells than in unperturbed cells.

## Global changes in protein levels in SAC-active cells

A persistent checkpoint activity represents a stressful condition. Recently, it was shown that large cell size per se also creates a situation of stress (28). Hence, SAC-active cells may elicit a response

at the transcriptional and translational level in analogy with the environmental stress response (ESR), a gene expression program that is activated in yeast under a wide variety of stressful conditions (29). We thus analyzed protein expression levels by performing an unbiased, high-throughput liquid chromatography tandem mass spectrometry (LC–MS/MS) analysis in cells where the checkpoint was induced either with Mad2 overexpression or the *tub2-401* allele.

The proteome of *tub2-401* cells dividing with an active SAC was compared with that of wild-type unperturbed cells at 22°C after 20 h. Instead, Mad2-overexpressing cells were compared with cells that express exogenous Mad2 but also a dominant negative form of Cdc20 (Cdc20-127) from the *tetO₂* promoter for 19 h. Cdc20-127 is not recognized by Mad2 (30), and its expression elicits the same effect as deleting the essential checkpoint gene *MAD3* (compare un-perturbed and *GAL1-MAD2 mad3Δ* in Fig S6A).

First, we asked whether cells elicit a common biological response in spite of the different ways in which the SAC was induced. To this aim, we analyzed the fold-change expressions of the two different conditions and found them to be significantly correlated. The correlation is even higher if we restrict the comparison to highly up- or down-regulated proteins (Fig 4A). In agreement with this result, principal component analysis (PCA), including all biological and technical replicates, reveals that SAC-active cells exhibit consistent changes, irrespective of the experimental condition (see PC2 in Fig S6B). This indicates that the two different ways of inducing the checkpoint trigger a common biological response.

To test whether the common traits of cells dividing with an active SAC are related to the ESR, we looked at changes in expression of

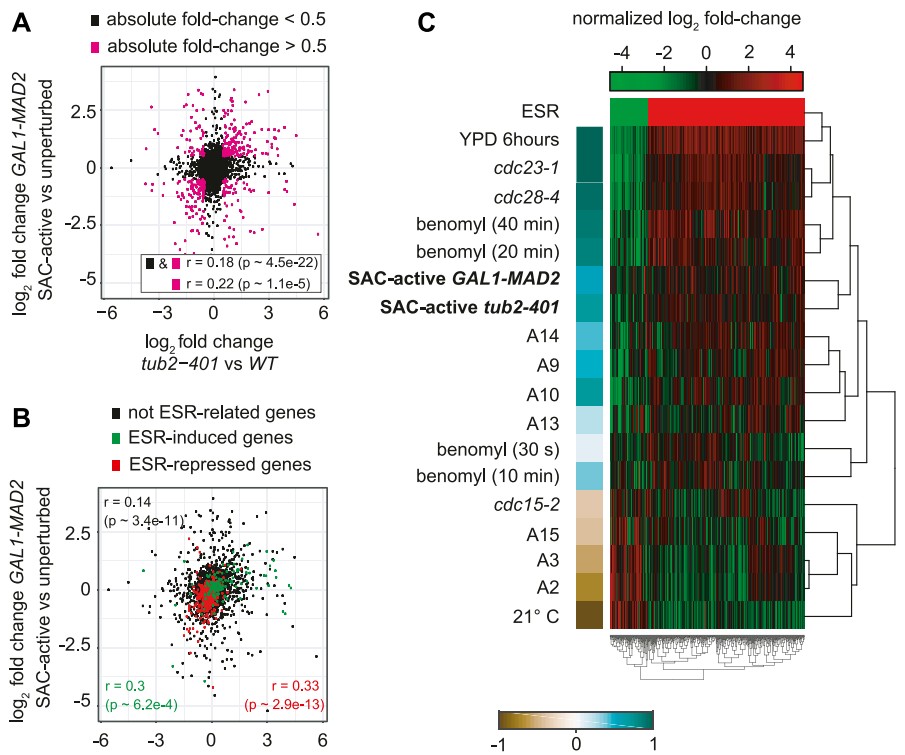

**Figure 4. The proteome of SAC-active cells undergoes global changes, which overlap with the ESR.**
**(A)** Scatter plot of log2-fold changes showing the common expression pattern of *GAL1-MAD2* and *tub2-401*. Proteins expressed at a log2-fold change above 0.5 are highlighted. Pearson correlation is calculated for all proteins and the highlighted subset, respectively. **(B)** Same as (A), highlighting proteins that are related to the ESR. Pearson correlation is separately calculated for proteins induced in, repressed in, and unrelated to the ESR, respectively. **(C)** Heatmap showing hierarchical clustering of protein expression in SAC-active cells and gene expression in other yeast strains: (i) early and late time points of cells upon benomyl treatment (31); (ii) the APC/C mutant *cdc23-1*; (iii) a mutant of CDK1, *cdc28-4*, with low activity in the S- and M-phase (32); (iv) a set of aneuploid cells (A2, A3, A9, A10, A13, A14, and A15) (33); (v) cells in stationary phase which show a very strong ESR expression profile (YPD 6 h) (29); (vi) data that do not show an ESR (*cdc15-2* cells (32)) and (vii) data which show an expression profile opposite to ESR (cells grown at low temperature (29)). The column to the left shows the normalized stress response intensity (SRI) that quantifies the presence of an ESR response in a sample—see the Materials and Methods section and Table S1 for details.

genes that have been identified as either induced or repressed in the ESR (29). In both cases, we observed a significant correlation of fold changes (Fig 4B). To confirm that the proteins that are differentially expressed in SAC-active cells are common to other conditions known to activate the ESR, we performed a cluster analysis, which includes our mass-spec data and data from various microarray experiments investigating the ESR response. We observe that cells with an active checkpoint cluster with cells that exhibit a strong ESR (stationary phase, *cdc23-1*, *cdc28-4*, and late time points of benomyl treatment, Fig 4C). Interestingly, SAC-active cells do not cluster with all aneuploid strains, which may reflect the fact that not all our cells are expected to be aneuploid (Fig S1D). The presence of an ESR in our samples is quantitatively confirmed by calculating the "Stress Response Intensity" (SRI) (33) which is also shown in Fig 4C and Table S1.

We then performed a gene ontology (GO) enrichment analysis of the proteome in cells dividing with an active checkpoint, distinguishing between up- and down-regulated proteins. "Cytoplasmic translation" was by far the most significant term among down-regulated proteins, and we found several other significant biological processes related to translation (Fig S6C), in line with the observation that growth is saturated in SAC-active cells. Among the up-regulated proteins, we found GO terms related to oxidative-stress—in agreement with ESR—but also unique terms (eg, "tricarboxylic acid cycle" and "drug transmembrane transport", Fig S6D).

In summary, we found that a sizeable fraction of the proteome is commonly regulated in cells where the checkpoint was induced with two different experimental protocols. Changes in protein expression that are typical of SAC-active cells are largely shared with the ESR.

## The SAC-active state is reversible

The observed phenotypic changes (size, proteome, and altered cell cycle) in SAC-active cells appear like a differentiated state that is markedly different from the unperturbed one. Thus, one may ask whether cells have been permanently altered or whether the changes are reversible. All the results presented so far are consistent with the hypothesis that the behavior of cells dividing under checkpoint-activating stimuli is a result of their prolonged mitosis. If so, as our model suggests, the SAC-active state could be reversed by lifting the checkpoint stimulus (Fig 5A).

We thus tested this prediction experimentally. Unlike altering the levels of Mad2 or Cdc20-127, Tub2 polymerization can be easily modulated by changing temperature in the *tub2-401* strain (Fig S1A). Hence, starting from *tub2-401* SAC-active cells, we increased the temperature to allow proper Tub2 polymerization (withdraw, in Fig 5B). We kept track of both cell number and cell size. As controls, we used *tub2-401* cells always grown at either restrictive (SAC-active) or permissive temperature (unperturbed).

Our results show that, after a transient of 3–4 h, the distribution of cell sizes overlaps with that of unperturbed cells (Fig 5C). Of note, during the transition, the distribution shows only one mode, suggesting that the whole population on average changes size and that there is no selection for a SAC-resistant subpopulation of cells. The size of cells after withdrawal diverges from SAC-active cells and approaches that of unperturbed cells. SAC-active and unperturbed cells, by contrast, do not change their size throughout the 9 h of the experiment (Fig 5D). In agreement with the size dynamics, cell growth after withdraw approaches that of unperturbed cells (Fig 5E).

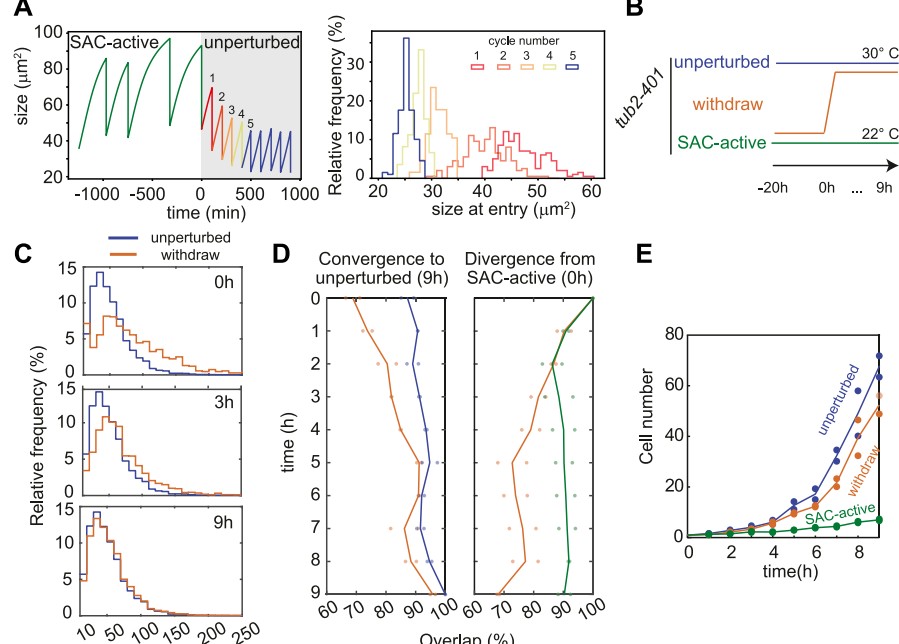

**Figure 5. Cell size and proliferative status of SAC-active cells are reversible.**
**(A)** Left: simulation of the withdrawal experiment. At time = 0 the distribution from which the cell cycle durations are drawn changes from SAC active to unperturbed. Right: Histograms of the size at entry (i.e., right after division) of the first five cycles after withdrawal. Each histogram corresponds to 200 individual simulations. **(B)** Experimental setup of the withdrawal experiment. *tub2-401* cells (yAC4096) grew in three different conditions: unperturbed at 30°C, SAC-active at 22°C, and withdraw were shifted to 30°C after 20 h at 22°C. Cells were monitored every hour for 9 h after the temperature shift. **(C)** Cell size distributions of unperturbed and cells after withdraw, measured at 0 h, 3 h, and 9 h after the temperature shift. **(D)** Overlap of cell size distribution after withdraw compared with unperturbed and SAC-active cells during the entire experiment—see the Material and Methods section for details. On the left: evaluation of convergence to unperturbed distribution at 9 h; on the right: evaluation of divergence from SAC-active distribution at 0 h. Individual dots represent biological replicates, whereas continuous lines the mean behaviors. **(E)** Growth curves of unperturbed, SAC-active, and withdrawn cells. Individual dots represent biological replicates, whereas continuous lines the mean behaviors.

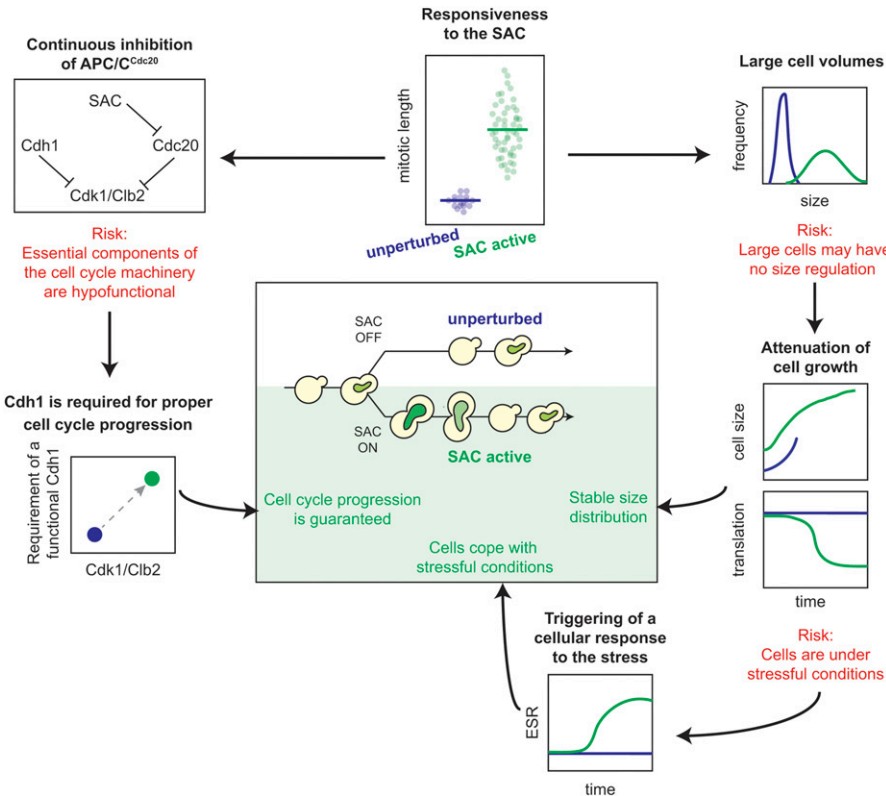

**Figure 6. Cellular responses supporting proliferation in the presence of an active mitotic checkpoint.** Schematic of the behavior of SAC-active cells. Under constant checkpoint activity, SAC-active cells are exposed to several risks and stressful conditions, to which they respond with appropriate strategies.

In conclusion, our data show that the dramatic changes induced by constant checkpoint activation are reversible, provided that the checkpoint is lifted after a few cycles.

## Conclusions

### Stable phenotypic properties of cells dividing under checkpoint conditions

We asked whether cells that proliferate under constant mitotic checkpoint activation are characterized by typical, stable properties. Alternatively, they could be an ensemble of many unique complex phenotypes driven by the different patterns of missegregation. We found that SAC-active cells have specific and reproducible behaviors, quite different from those of unperturbed cells. The first and most important property is that SAC-active cells proliferate while at the same time mounting a checkpoint response during each cell cycle. Unlike prolonged treatment with α-factor (15), cells that escape the arrest do not preserve a memory of this event, but each time mount a proper checkpoint response.

The mitotic checkpoint targets an essential cell cycle component, Cdc20. Not surprisingly, cell proliferation is greatly disturbed when Cdc20 is constantly inhibited. Yet, cells show a great deal of plasticity and manage to divide efficiently, albeit more slowly, by rebalancing their growth and division. All the phenotypes we have identified can be understood in this light (Fig 6). The longer

mitoses come with larger sizes of SAC-active cells. Although cells become bigger, the distribution of cell sizes is stable because of the saturation of growth rate observed in cells delayed in mitosis. Saturation of growth and the attainment of a larger size correlate with changes in the expression levels of proteins involved in transcription and translation. Other proteins whose levels are changed in SAC-active cells compared with unperturbed cells are shared with the ESR, a program of gene expression that is triggered by many stressful conditions. Finally, the inhibition of Cdc20 requires cells to rely more heavily on Cdh1, the other activator of APC/C.

This latter result is in line with the idea that cells can either cycle with a Cdc20- or a Cdh1-driven cell cycle oscillator, the second becoming essential only when the first is impaired (20). Either one or the other are needed to keep CDK1 activity under control (19, 22). This is true in particular in SAC-active cells, where we previously showed that overcoming the checkpoint requires CDK1 activity and that even minimal inhibition of CDK1 may prevent cells from overcoming the arrest (9). Thus, in cells proliferating with an active SAC, it is particularly important to reverse the high CDK1 activity required to enter anaphase. Here lies the role of *CDH1*, whose deletion impairs cells from exiting mitosis and starting a new cycle, as suggested by the negative synthetic interaction with checkpoint activation. In this interpretation, Cdh1 is only required after the transition into anaphase, which in SAC-active cells is driven by few molecules of APC/C^Cdc20 escaping inhibition from the checkpoint (as proposed in reference 10). We hypothesize that Cdh1 is not the only element to play a more prominent role in the cycle of

SAC-active cells after anaphase onset because cells manage to divide a few times even without *CDH1*, with high levels of Clb2. Hence, the phosphatases opposing Cdk1 (in budding yeast, Cdc14 (34)) as well as Cdk1 inhibitors such as Sic1 may also play an important role during proliferation of cells dividing with an active checkpoint (22).

### SAC-active cells show a unique phenotype

Taken one by one, the properties of SAC-active cells are shared with other mutants. The global profile of protein expression resembles that of cells that express APC/C mutants (*cdc23-1*) or which have low kinase activity and are delayed in the cell cycle (*cdc28*-4) (32). Also, they share many elements with the ESR (29), which is also activated by aneuploidy (32). Growth of SAC-active cells saturates in mitosis, similarly to that of cells arrested in mitosis by Cdc20 deprivation (26). Large cells have recently been reported to saturate growth (28). However, none of these conditions alone recapitulates the properties of cells dividing upon checkpoint-activating conditions. For example, the *cdc23-1* APC/C mutant is not reported to be impaired in chromosome segregation (32); depletion of Cdc20 arrests cells permanently before anaphase (35); aneuploidy does not come with APC/C inactivation and prolonged mitosis (32, 33); and in our system, growth saturates for sizes that are smaller (roughly 50%) than that observed in the G1-arrested cells described in (28).

Hence, the ensemble of phenotypes we have identified represents a unique and specific trait of SAC-active cells whose molecular characterization will be an important future goal.

### The beginning of a relay race

It has been proposed that cells react to external stimuli via a relay race that starts with changes in protein expression, which occur in the short-term period, and ends with genetic alterations, which induce irreversible, long-term modifications (36). Within this framework, our data depict the quick reaction (2–3 cycles) to a persistent mitotic block. At later time points, cells will acquire mutations that cause resistance to antimitotics by preventing them from altering tubulin dynamics. When this happens and cells are no longer delayed in mitosis, they are likely to lose many of the properties we have described, as suggested by the fact that cells revert their phenotype after removing the checkpoint signal (Fig 5). Thus, we speculate that there exists a window of opportunity for selectively attacking cells that manage to overcome a checkpoint arrest and yet have not become genetically resistant to the drugs. Our results, showing that cells seem to react in a similar fashion to different checkpoint-activating stimuli, suggest that the strategy can be quite effective in targeting most SAC-active cells. Moreover, some of the results observed in yeast may suggest ways to operate in mammalian cells. For example, the fact that SAC-active cells become sensitive to Cdh1 deletion suggests that Cdc20 should not be the only target of antimitotics, but a double inhibition of Cdc20 and Cdh1 may elicit better results. Further analysis of our mass-spec data, and especially of the proteins specifically altered in SAC-active cells, may offer additional entry points.

## Materials and Methods

### Strains

All strains are listed in Table S2 and were derivative of or were backcrossed at least three times with W303 (*ade2-1*, *trp1-1*, *leu2-3,112*, *his3-11*, and *ura3*). Original construct for *GAL1-MAD2* was developed in the laboratory of S Piatti (Centre de Recherche en Biologie Cellulaire de Montpellier, Montpellier, France); *SIC1(10x)* was received by DP Toczyski (Department of Biochemistry, University of California, San Francisco, CA, USA). *CLB2-GFP* was received by PA Silver (Department of Biological Chemistry and Molecular Pharmacology, Harvard Medical School and The Dana-Farber Cancer Institute, Boston, MA, USA). *TUB2-mCherry* and *cdh1Δ* strains were received by R Visintin (Department of Experimental Oncology, European Institute of Oncology, Milan, Italy). *HTB2-mCherry* was received by FR Cross (The Rockefeller University, New York, NY, USA). *ChrV-GFP* (*his3-11,15::HIS3tetR-GFP ura3::3xURA3tetO_{112}*) was received from S. Piatti.

*tetO_2-CDC20-127* cells were obtained by yeast transformation of wild-type strain yAC1001 with plasmid pBS94—see Table S3—from A Murray (Department of Molecular and Cellular Biology, Harvard University, Cambridge, MA, USA), carrying a copy of *CDC20-127* (30) with the *TRP1* marker under the doxycycline-repressible promoter *tetO_2*. The plasmid was digested with three different restriction enzymes (EcoRV, Bsu36I, and BstNI). Transformants were tested for single insertion by Southern blot, whereas bypass of the mitotic arrest induced by nocodazole 15 μg/ml was tested by FACS analysis.

*tub2-401* cells were obtained by transformation of the wild-type strain yAC1001 with the plasmid pTH18 (see Table S3) from T Huffaker (Department of Molecular Biology and Genetics, Cornell University, Ithaca, NY, USA), which contains a copy of *TUB2* ORF carrying the described four point mutations (11) with an *URA3* marker. The plasmid was digested with the restriction enzyme KpnI before transformation (11). Transformants were than plated on 5-fluoro-orotic acid plates to select against *URA3*, and the resulting colonies were amplified and tested for cold sensitivity at 15°C. *tub2-401* ORF was checked by Sanger sequencing.

### Media and reagents

All population experiments were performed using yeast extract peptone (YEP) medium (1% yeast extract, 2% Bacto Peptone, and 50 mg/l adenine) supplemented with 2% glucose (YEPD), 2% raffinose (YEPR), or 2% raffinose and 2% galactose (YEPRG). Live-cell imaging experiments were performed using synthetic complete (SC) medium, supplemented with ammonium sulfate and 2% glucose (SCD), 2% raffinose (SCR), or 2% raffinose and 2% galactose (SCRG) (10).

To synchronize cells in G1, α-factor (GenScript) was used at 5 μg/ml for 1 h 30 min, followed by 2.5 μg/ml for 30/45 min. In the case of *tub2-401* cells, an additional hour with α-factor 5 μg/ml was introduced before the release, while cells experienced the temperature shift. After the release from G1 in population experiments, re-addition of α-factor was performed at 20 μg/ml when more than 90% of the cells were budded, and then every 2 h. Doxycycline hyclate (Sigma-Aldrich) was used at 10 μg/ml according to the

method described by reference 37, to repress Cdc20-127 expression in *tetO₂-CDC20-127* cells.

In the case of *GAL1-MAD2* cells, galactose 2% was added 1 h before the release from α-factor. Except for *tub2-401* cells, all the experiments were performed at 30°C.

## Proteomic analysis

### Cell lysis

From a culture in the log-phase growth, 10 ml of cells were pelleted by centrifugation and resuspended in 1 ml of 100 mM Tris HCl, pH 7.6. The cells were pelleted again and subjected to rapid cooling in dry ice plus denatured alcohol and stored at −80° C overnight. Once thawed, the cells were resuspended in 80 μl of lysis buffer (100 mM Tris HCl, pH 7.6, 100 mM dithiothreitol, and 5% SDS) and incubated at 95°C for 5 min. Cell lysis was performed by adding glass beads in each sample and vortexing for 10 min. After an addition of 40 μl of lysis buffer, the cell lysate was transferred in a new tube and centrifuged at the maximum speed. Supernatant was collected and stored at −20°C.

### Protein digestion for MS analysis

Protein denaturation was performed by resuspending 50 μg of lysate in 200 μl of UA buffer (100 mM Tris HCl, pH 8.5, and 8 M urea) and transferring it in YM-30 micron filters (Millipore). Each lysate was pelleted and washed three times with UA buffer. The lysate was then reduced with 10 mM DTT in UA buffer for 30 min at room temperature. After two additional washes with UA buffer, each sample was incubated in 100 μl of 50 mM iodoacetamide and 8 M urea for protein alkylation. After two washes with UA buffer and two with 40 mM NH₄HCO₃, protein in-solution digestion was performed by incubating each sample in 95 μl of 40 mM NH₄HCO₃ supplemented with 120 mM CaCl₂ and 1 μg of trypsin overnight at 37°C. Then, an additional incubation with 1 μg of trypsin for 3 h was performed. The obtained peptides were collected by centrifugation, purified on a C18 StageTip (Proxeon Biosystems) and split in two independent samples for technical replicates.

### Mass spectrometry analysis

For each sample, 1 μg of peptides was injected in a Q-exactive HF mass spectrometer (Thermo Fisher Scientific). Peptides were separated on a linear gradient from 95% solvent A (2% acetonitrile and 0.1% formic acid) to 55% solvent B (80% acetonitrile and 0.1% formic acid) over 120 min, followed by 100% of solvent B in 3 min at a constant flow rate of 0.25 μl/min on ultra-high-performance liquid chromatograhphy (UHPLC) Easy-nLC 1000 (Thermo Fisher Scientific). The LC system was connected to a 23-cm fused silica emitter of 75-μm inner diameter (New Objective, Inc), packed in-house with ReproSil-Pur C18-AQ 1.9-μm beads (Dr Maisch Gmbh) using a high-pressure bomb loader (Proxeon). A data-dependent acquisition was performed with the following settings: enabled dynamic exclusion of 20 s, MS1 resolution of 60,000 at m/z 200, MS1 automatic gain control target of 3e+6, MS1 maximum fill time of 20 ms, MS2 resolution of 15,000, MS2 automatic gain control target of 1e+5, MS2 maximum fill time of 80 ms, and MS2 normalized collision energy of 28. For each data-dependent acquisition cycle, one full MS1 scan range of 300–1,650 m/z was followed by 12 MS2 scans, using an isolation window size of 2 m/z. The resulting proteomic data have been loaded into PeptideAtlas repository (Dataset Identifier PASS01302).

### Database search

Raw data coming from MS analysis were processed with MaxQuant software (1.5.6.0) (38), using Andromeda search engine (39). MS/MS peak lists were searched against the UniProtKB/Swiss-Prot protein sequence yeast complete proteome database (release 2014). A reverse decoy database was generated within Andromeda, setting a 0.01 false discovery rate for peptide spectrum matches and proteins. A filtering was applied to the resulting list, asking at least two peptides identifications per protein, of which at least one peptide had to be unique to the protein group. Proteins were analyzed in a label-free manner, using LFQ intensities, which represent protein intensity values normalized across the entire data set.

### Data analysis

Data analysis based on LFQ intensities was performed using the R Bioconductor Package "DEP" (40). The data were background-corrected and normalized by variance stabilizing transformation (vsn). Missing values were imputed using the quantile regression-based left-censored function ("QRILC"). PCA revealed strong batch effect for experiments performed on different days. Batch correction was performed using the R Bioconductor package "SVA" (41). Differential enrichment analysis was performed with the R Bioconductor package "limma" (42), applying linear models with a moderated *t* test statistic while taking into account the correlation of technical replicates. Gene ontology analysis was carried out with the R Bioconductor package "topGO" (43), testing enrichment with a Kolmogorov–Smirnov test on genes ranked by fold change and identifying GO terms using topGO's default method. Microarray data for the cluster analysis used in Fig 4 were retrieved from the Gene Expression Omnibus database. To make the data sets comparable across different experimental setups, all data sets (log2-fold-changes) were first normalized by using vsn before subtracting the mean and dividing by the SD. Clustering was then performed using the complete linkage method for agglomeration and Spearman's rank correlation as a distance measure (dist = 1 − cor). Applying other common distance measures yielded similar results. The normalized stress response intensity (SRI) was calculated in line with (32) as:

$$SRI = \frac{\sum_{i=1}^{m} x_i + \sum_{j=1}^{n} (-y_j)}{\sum_{i=1}^{m} |x_i| + \sum_{j=1}^{n} |y_j|},$$

where the $x_i$ and $y_i$ refer to the fold changes of genes that are induced and repressed in the ESR, respectively (33).

## Single-cell experiments

### General image acquisition settings

Each single-cell experiment was performed monitoring cells growing in microfluidic chambers (CellASIC), in which flowing

medium was maintained with an ONIX microfluidic perfusion system (CellASIC). Time-lapse movies were recorded using a DeltaVision Elite imaging system (Applied Precision) based on an inverted microscope (IX71; Olympus), a UPlanFL N 60× (1.25 NA) oil immersion objective lens (Olympus) and a camera (Scientific CMOS Camera). During experiments at 30°C, an oil immersion with refractive index n = 1.516 was used. In the case of experiments performed at the semipermissive temperature of 19–20°C, we used oil immersion with n = 1.512. GFP and mCherry were acquired using single bandpass filters (EX475/28 EM523/36 for GFP and EX575/25 EM632/60 for mCherry). Excited and nonexcited fields were acquired, to evaluate any phototoxicity of the acquisition settings by comparing the cell cycle duration in excited and nonexcited cells. For live-cell imaging experiments, the following settings were used: Clb2-GFP with 1 z-stack, exposure time 0.15 s, and power lamp 32%; Tub2-mCherry with 3 z-stacks spaced 0.85 $\mu m$, exposure time 0.10 s, and power lamp 10%; and Htb2-mCherry with 1 z-stack, exposure time 0.7 s, and power lamp 10%. Frames were taken every 10 min for *GAL1-MAD2* cells, whereas every 15 min for *tub2-401* cells. For samples fixed in cold EtOH (Fig S1D), the following settings were used: ChrV-GFP with 35 z-stacks spaced 0.2 $\mu m$, exposure time 0.1 s, and power lamp 32%; Htb2-mcherry with 35 z-stacks spaced 0.2 $\mu m$, exposure time 0.1 s, and power lamp 10%. For immunofluorescence (Fig S1A), the following settings were used: DAPI with 11 z-stacks spaced 0.2 $\mu m$, exposure time 0.2 s, and power lamp 50%; FITC with 11 z-stacks spaced 0.2 $\mu m$, exposure time 0.025 s, and power lamp 10%. In the case of Tub2-mCherry, a maximum intensity projection among the z-stacks was performed with Fiji. In the case of samples fixed in cold EtOH and immunofluorescence, z-stacks were deconvolved with SoftWoRx software and then projected with maximum intensity projection with Fiji.

### Image acquisition settings at low temperatures
Experiments with *tub2-401* cells were performed at the semi-permissive temperature of 19–20°C, as detected by the incubator, enveloping the microfluidic plate in a metallic chamber in which a refrigerated antifreeze mixture was flowing. Temperature control was performed by a BOLD LINE Water-Jacket Top Stage Incubation System (OKOlab). In this system, refrigeration and flow of antifreeze was performed by an immersion thermostat (LAUDA DR. R. WOBSER GMBH & CO. KG, Germany), which was connected to the incubating chamber with several insulated tubes. Moreover, an objective cooler was connected to the same system, to refrigerate the objective during the time lapse. Temperature control was performed by BOLD LINE T-unit, together with SmartBox (OKOlab). The system took into account the room temperature—measured by a thermistor—and the temperature inside the incubating chamber—monitored by a fine gauge thermocouple. T-unit allowed two ways of temperature control: Sample Mode (keeping constant the temperature of the specimen monitored by the thermocouple) or Chamber Mode (maintaining a constant temperature of the chamber). A BOLD LINE Logger Software (OKOlab) allowed communications with OKOlab system to log data from it.

### Segmentation, tracking, and data extraction
Each time lapse was imported in Phylocell, an open-source software written in MATLAB by G. Charvin (Institut de Génétique et Biologie

Moléculaire et Cellulaire, Illkirch, Graffenstaden, France) and available at https://github.com/gcharvin/phyloCell. Using Phylocell, segmentation of cell bodies was achieved with homothetic inflation and/or watershed algorithms, manually adjusting the obtained areas where necessary. Tracking of the segmented cells in time was performed by the Iterative Closest Point algorithm. From each segmented cell, we extracted cellular area and fluorescence in time, using ad-hoc software written in MATLAB. During data extraction, background fluorescence was subtracted. Mother and daughter cells were considered as a unique region of interest for each cell cycle—that is, from budding event until the rebudding of mother cell.

### Measuring cell cycle and mitotic duration
For simplifying the analysis, we followed only the progeny of cells present at the beginning of the time lapse. Evaluation of the mean nuclear Clb2-GFP signal for the entire cell cycle was implemented with the k-means algorithm, as described in (9). Each Clb2-GFP trajectory was then smoothed with a Savitzky–Golay filter and plotted, to check one by one the behavior of each cell cycle with a basic user interface. Selection of the mitotic entry was defined by the user at the time in which Clb2 started to increase, whereas anaphase when Clb2 started to decrease. Mitotic length was defined as the difference between these two timepoints. Cell cycle length was defined as the time between two subsequent budding events. Fold-increases of mitotic and nonmitotic phases were evaluated with the following ratios:

$$FOLD_{mitosis} = \frac{m_{mitosis}^{SAC-active}}{m_{mitosis}^{unperturbed}}$$

$$FOLD_{not\,mitosis} = \frac{m_{cell\,cycle}^{SAC-active} - m_{mitosis}^{SAC-active}}{m_{cell\,cycle}^{unperturbed} - m_{mitosis}^{unperturbed}},$$

where $m_{mitosis}^{unperturbed}$ and $m_{mitosis}^{SAC-active}$ represent the median mitotic length in unperturbed and SAC-active cells, whereas $m_{cell\,cycle}^{unperturbed}$ and $m_{cell\,cycle}^{SAC-active}$ the median cell cycle length in unperturbed and SAC-active cells. In the case of experiments with *cdh1Δ* cells, nuclear segregation was the readout of anaphase onset. The event was identified by looking at the separation of histone Htb2-mCherry masses; in this case, mitotic length was defined as the time interval between budding and nuclear segregation.

### Cell size analysis
Cellular area was used as a proxy of cell size, by converting the number of pixels in $\mu m^2$ (1 pixel ~ 0.05 $\mu m^2$). Size of gen0 unperturbed and gen0 SAC-active cells was monitored from G1 release until the beginning of Clb2 degradation, while unperturbed and SAC-active cells were monitored from budding to the beginning of Clb2 degradation. Each size trajectory was filtered for noisy data, mostly because of errors in the segmentation process and/or to an altered phase contrast during the time lapse—see next section, "Filtering and smoothing of cell size trajectories." For each cell population, the mean size trajectory was evaluated in time, by considering only those timepoints in which more than five size trajectories were available (Fig 3C). Finally, the local growth rate was evaluated as the first derivative of each smoothed size

trajectory, by using a central finite differences method. Values of local growth rate were plotted and binned respect to the cell size (window of the bins: 10 $\mu m^2$) and mean values of cells size and local growth rate were evaluated for each bin (Fig S5B, squares). A similar analysis was performed by using Clb2 levels in abscissa (Fig S5A), with bins having a window of 70 a.u./pixel for *GAL1-MAD2* and 100 a.u./pixel for *tub2-401* cells.

### Filtering and smoothing of cell size trajectories

For each point of a size trajectory,

$$\overline{A} = [A_i]_{i=1}^{N} = [A_1, A_2, ..., A_N],$$

the corresponding vector of the local size variations was evaluated as

$$\Delta\overline{A} = [A_{i+1} - A_i]_{i=1}^{N-1} = [A_2 - A_1, \ A_3 - A_2, ..., \ A_N - A_{N-1}].$$

For each size trajectory of gen0 unperturbed, unperturbed, gen0 SAC-active, and SAC-active cells, local size variations $\Delta A = A_{i+1} - A_i$ were plotted respect to the size values $A_i$ (Fig S4D) and then clustered in three groups by the corresponding size values:

$$\Delta\overline{A}_{0-40} = \left\{ A_{i+1} - A_i \in \Delta\overline{A} \mid A_i \leq 40 \ \mu m^2 \right\}$$
$$\Delta\overline{A}_{40-80} = \left\{ A_{i+1} - A_i \in \Delta\overline{A} \mid 40 \ \mu m^2 < A_i \leq 80 \ \mu m^2 \right\}$$
$$\Delta\overline{A}_{80-120} = \left\{ A_{i+1} - A_i \in \Delta\overline{A} \mid 80 \ \mu m^2 < A_i \leq 120 \ \mu m^2 \right\}.$$

Some of the data were adjusted according to the following procedure. We identified values $A_i$ whose local variation $\Delta A$ was below 1.5 times the interquartile range of the first quartile or above 1.5 times the interquartile range the third quartile of the group distribution (Fig S4D, data outside the green boxes). These data $A_i$ were then replaced by the mean value of the two adjacent data (Fig S4E, red and green points). The process was performed for at most 10 iterations. Once adjusted, each size trajectory was smoothed with a moving average filter (Fig S4F) with the following windows: three timepoints for gen0 unperturbed and unperturbed cells, eight timepoints for gen0 SAC-active and SAC-active cells.

### Evaluation of mis-segregation events

For each strain reported in Fig S1D, 1 ml of cells was pelleted, fixed in cold EtOH 100% and stored it at −20°C. Then, 200 $\mu l$ of fixed cells were resuspended in 800 $\mu l$ of 50 mM Tris, pH 7.6. The cells were sonicated, pelleted, and resuspended in the remaining liquid after discarding the supernatant. 5 $\mu l$ of cells were loaded on a slide coated with 2% agar layer and then imaged for ChrV-GFP and Htb2-mCherry as reported in the Materials and Methods section—"General image acquisition settings." Cells were classified as mis-segregating if they satisfied one of the following requirements: (i) presence of >2 GFP foci in one cellular body and (ii) 2 GFP foci and two distinct Htb2 masses (i.e., nuclear masses) in one cellular body. The rate $R$ of ChrV mis-segregation was evaluated for each condition (Fig S1D), by normalizing the number of mis-segregating cells to the total number of scored cells. Assuming that each chromosome mis-segregates with the same frequency, the probability $P$ of

having at least one aneuploid chromosome among the total of 16 chromosomes of *S. cerevisiae* is given by

$$P = \mathbb{P}(\# \ aneuploid \ chromosomes \geq 1) =$$
$$1 - \mathbb{P}(\# \ aneuploid \ chromosomes = 0) =$$

$$= 1 - \prod_{i=1}^{16} \mathbb{P}(Chromosome \ i \ does \ not \ missegregate) =$$

$$= 1 - \prod_{i=1}^{16} [1 - \mathbb{P}(Chromosome \ i \ missegregate)] =$$

$$= 1 - (1 - R)^{16},$$

where $R$ represents the experimental rate of ChrV mis-segregation—see Fig S1D.

### Statistical analysis

Sample size was not evaluated a-priori for each experiment. Representation of dotplot distributions and growth curves was performed with Prism (GraphPad software), whereas all the other plots and statistical analyses were performed with MATLAB (MathWorks), using a significance level $\alpha = 0.05$. Normality of each distribution was verified with the following tests: Lilliefors' composite, Anderson-Darling, Jarque-Bera and single-sample Kolmogorov–Smirnov; the normality hypothesis was not rejected when at least three of four tests returned a $P$-value greater than the significance level. The comparison of two normal distributions was performed by a two-tailed unpaired *t* test with Welch's correction; in the case of non-normal distributions, a two-tailed Mann–Whitney test was used. Comparison of the cell cycle length, mitotic length, and anaphase-to-G1 length between two populations was performed with a log-rank test, to take into account the presence of censored data—that is, cells that did not end up with the observed cell cycle phase because of cellular death or to a precocious end of cell monitoring. Linear dependency between two datasets was evaluated either by measuring Pearson or Spearman correlation coefficients—according to the normality of the distributions—or by performing linear regression on data—considering the adjusted $R^2$ as a measure of goodness of fit and evaluating the significance of the estimated slope. Distributions in Fig S4A were evaluated with a kernel density estimator. In Fig 5D, the overlap O of two size probability density functions $f$ and $g$ was evaluated in the following way:

$$O(f, g) = \int_{10 \ fL}^{3050 \ fL} \min[f(v), g(v)] \ dv.$$

The following symbols were used for summarizing $P$-values: NS ($P > 0.05$), * ($P < 0.05$), ** ($P < 0.01$), *** ($P < 0.001$), and **** ($P < 0.0001$).

### Other techniques

### FACS analysis

In this study, FACS analysis was used for an evaluation of DNA content in a population of cells (see Fig S6A). For each timepoint, 1 ml of cells was collected and fixed with 1 ml of ethanol 70%. After

incubation with RNAse 1 mg/ml (Sigma-Aldrich) for 4–5 h or during an entire overnight, DNA was stained by propidium iodide 50 $\mu$g/ml. After a proper sonication, the stained cells were processed by flow cytometry (FACScalibur, DB), scoring 10,000 event for each sample. Raw data were then analyzed with FlowJo software.

### Serial dilutions and drop test assay

Cells were grown overnight in 5 ml of liquid medium, allowing them to reach a high concentration during the overnight incubation. After diluting all of them to the same concentration, the cells were further diluted 1:10 in sterile water and 200 $\mu$l of diluted cells were loaded on the first column of a 96 well. 180 $\mu$l of sterile water were added from the second to the sixth column. Then, 20 $\mu$l of cells from the first column were collected, and serial dilution was performed from the second to the sixth column. Finally, cells were spotted on agar plates with a mold or multichannel pipette.

### Immunofluorescence

This technique was used for visualization of the spindles in *WT* and *tub2-401* cells (Fig S1A). For this purpose, 1 ml of cells was pelleted and resuspended with 1 ml of KPi buffer (0.1 M Kphos, pH 6.4—obtained by mixing $K_2HPO_4$ with $KH_2PO_4$—and 0.5 mM $MgCl_2$) supplemented with 3.7% formaldehyde to fix cells. After three washes with KPi buffer and one with sorbitol solution (Sorbitol 1.2 M, 0.1 M Kphos, pH 6.4, and 0.5 mM $MgCl_2$), each sample was incubated at 37°C in 200 $\mu$l of mix solution (sorbitol solution supplemented with zymolase 10 mg/ml and 2-mercaptoethanol 0.2%), to allow cell wall digestion. When spheroplasts were visible, they were collected by centrifugation and washed with sorbitol solution. The resulting pellets were stored at -20°C. Once thawed, 5 $\mu$l of spheroplasts were loaded into glass slides (Thermo Fisher Scientific) coated with poly-L-lysine solution (Sigma-Aldrich). After 15/20 min, the slides were immersed in cold methanol (–20°C) for 3 min and then in cold acetone (–20°C) for 10 s, to dehydrate and fix samples. Each sample was incubated with anti-Tub1 primary antibody (MCA78G; Bio-Rad) and FITC-conjugated anti-rat secondary antibody (Jackson ImmunoResearch Laboratories). Finally, staining of DNA was performed with DAPI 50 ng/ml and slides were closed.

### Growth curves and cell size measurements in population experiments

Cellular concentration and size distribution of a log-phase culture were determined by using Scepter Handheld Automated Cell Counter (Merck Millipore), with 40 $\mu$m Scepter Cell Counter Sensors (Merck Millipore). 1 ml of yeast culture was sampled, sonicated, and diluted 1:100 in a final volume of 500 $\mu$l of PBS(1×). The diluted cells were vortexed and processed at the Cell Counter. Measured values were then exported with Scepter Software Pro (Merck Millipore). In Fig 5E, cellular concentrations are normalized with respect to the concentration at time 0.

### Mathematical modelling of cell size in unperturbed and SAC-active cells

Our model is based on a logistic growth function multiplied to a linear term:

$$A(t) = \frac{A_0 A_{max} e^{rt}}{A_{max} + A_0(e^{rt} - 1)} \left(1 + \frac{m}{A_{max}} t\right).$$

The resulting growth curve is exponential ($\sim e^{rt}$) for small $t$, and for large $t$ converges to a straight line with slope $m$. $A_0$ refers to the

size at $t = 0$ and $A_{max}$ determines the size at which the transition between exponential and linear behavior occurs. In rough agreement with the measured data, $A_0$ and $A_{max}$ were chosen as 20 and 120 $\mu m^2$, respectively. Cycle lengths for unperturbed cells were drawn from a normal distribution with a mean of 100 min and a SD of 10 min. Cycle lengths for G1 SAC-active and SAC-active cells were drawn from an exponential distribution with a mean of 200 min to which a delay of 200 min was added. Using these numbers and assuming that the unperturbed cells double in size (i.e., $A(T_{unperturbed}) = A(100\ min) = 2A_0$), the growth parameter was calculated as follows:

$$r = \frac{log(2(A_{max} - A_0)) - log(A_{max} - 2A_0)}{T_{unperturbed}} \approx 0.0092.$$

Finally, cell division was modeled by instantaneously dividing cell size by 2.

### Biological and technical replicates

Data were representative of two biological replicates. In the case of LC–MS/MS analysis, two biological replicates for *GAL1-MAD2* and three biological replicates for *tub2-401* were analyzed: for each biological replicate, two technical replicates were included in the analysis.

# Supplementary Information

# Acknowledgements

We thank Silke Hauf, Jelena Vermezovic, Adrian Saurin, Dávid Szűts, and Andrea Musacchio for comments on the manuscript; Luca Mariani for the characterization of the Cdc20-127 mutant; Paolo Bonaiuti for customization of data extraction from single-cell movies; Mattia Pavani for helping at different stages of the project; Georg Holtermann for support with imaging at low temperatures; and Pierre-Luc Germain for suggestions on the analysis of mass-spec data. The lab of A Ciliberto is supported by the Associazione italiana ricerca sul cancro (AIRC) IG grant IG 17490 and the lab of A Bachi is supported by the AIRC IG 18607. A Corno is recipient of an AIRC-LoveDesign fellowship (project code 18188). We thank the IFOM Imaging Facility, and particularly Sara Barozzi and Amanda Oldani, for their support especially in the *tub2-401* movies, and Emanuele Martini for image analysis. A Ciliberto thanks the Hungarian Academy of Sciences and in particular the Institute of Enzymology, for hospitality.

## Author Contributions

A Corno: conceptualization, data curation, formal analysis, investigation, and writing—review and editing.
E Chiroli: data curation and investigation.
F Gross: data curation, formal analysis, investigation, and writing—review and editing.
C Vernieri: investigation and writing—review and editing.
V Matafora: data curation and formal analysis.
S Maffini: methodology.

M Cosentino Lagomarsino: methodology.

A Bachi: formal analysis, methodology, and writing—review and editing.

A Ciliberto: conceptualization, supervision, funding acquisition, project administration, and writing—original draft, review, and editing.

**Conflict of Interest Statement**

The authors declare that they have no conflict of interest.

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
