## [Reviewer comments · Life Science Alliance]

Life Science Alliance

Cellular response upon proliferation in the presence of an active mitotic checkpoint

Andrea Corno, Elena Chiroli, Fridolin Gross, Claudio Vernieri, Vittoria Matafora, Stefano Maffini, Marco Cosentino Lagomarsino, Angela Bachi, and Andrea Ciliberto

DOI: <https://doi.org/10.26508/lsa.201900380>

Corresponding author(s): Andrea Ciliberto, IFOM, Fondazione Istituto FIRC di Oncologia Molecolare

Review Timeline:

Submission Date:	2019-03-13
Editorial Decision:	2019-03-13
Revision Received:	2019-04-18
Editorial Decision:	2019-04-25
Revision Received:	2019-04-26
Accepted:	2019-04-29

Scientific Editor: Andrea Leibfried

Transaction Report:

Please note that the manuscript was previously reviewed at another journal and the reports were taken into account in the decision-making process at Life Science Alliance.

Referee #1 Review

Report for Author:

The paper concerns the phenomenon of mitotic 'slippage', where cells with an activated spindle assembly checkpoint (SAC) nevertheless proceed into mitosis. Two means of activating the SAC were employed: 'gratuitous' induction by overexpressing Mad2 in the absence of significant spindle perturbation; and partial spindle dysfunction with the tub2-401 cs mutation at semipermissive temperature. In general, results are considered relevant to slippage if found in both systems. This is a good approach that bypasses the substantial difference between the approaches. There could still be an issue that neither approach is necessarily equivalent to complete checkpoint activation; tub2-401 cells are viable under these conditions so most spindles are fully functional most of the time (so at least some of the cycles observed are in fact normal cycles - thus no slippage required!).

In contrast, Mad2 overexpression does nothing much to the spindle (though it is not inert, as shown here), so if a given level of Mad2 fully activates the checkpoint, that makes for easy interpretation. But do we really know this? After all, the normal checkpoint DOESN'T work by increasing Mad2 expression, rather by converting protein-protein interactions of Mad2 and the host of other kinetochore/MCC proteins. However, as noted, the use of both approaches does deal with these issues reasonably well.

What is learned? (1) Clb2 (mitotic cyclin) rises to high levels during the checkpoint arrests, and declines sharply during slippage exits. This is consistent with previous work in various systems. It is shown here that this pattern repeats in subsequent cycles, which is interesting to know (while the well-known necessity for B-cyclin-CDK activity to drop to allow mitotic exit makes it a reasonably likely result going in, it was possible that in later cycles cells might somehow 'discover' some other means, such as SIC1 overexpression). This finding indicates that slipping cells are not stably 'adapted' to a continued checkpoint signal. (2) Slippage is at least somewhat dependent on CDH1, an alternate APC activator to CDC20 (the target of SAC inhibition). This is interesting; essentiality of CDH1 in cases where CDC20 is compromised has been reported in various contexts (including in mitotic slippage in animal cells). Ref. 9 (from the authors' laboratory) showed no requirement for CDH1 for the first anaphase in this same system with GAL-MAD2 (at least, for loss of a scored mitotic spindle). What is shown here is that subsequent mitotic exit is considerably slower without CDH1. I guess this suggests incomplete CDC20 activation, enough to get at securin but not enough to clear CLB2 low enough to allow mitotic exit without assistance from CDH1 (this is my own interpretation because I couldn't exactly get the authors' ideas here). This should all be explained more clearly because as I understand it, this is a very different perspective on the role of CDH1 in slippage from that proposed in ref. 9 (where the simple statement was that CDH1 was not involved). (3) There is some examination of cell size under these conditions. The checkpoint-delayed cells are larger than unperturbed cells, which is completely expected (since cell growth continues for some time at cell cycle blocks, as first shown by Hartwell in the 1970's). It is further shown that cell growth rate slows as cells undergo prolonged checkpoint delays. This is potentially interesting but needs to be compared to growth rates of similar-sized cells due to different blocks, in order to disentangle a checkpoint-specific response from truly huge cells slowing down because they only have one nucleus, only have one (or two) genome equivalents, have a low surface-to-volume ratio, etc etc. (4) The SAC-active cells induce a proteomic response with overlap to a general stress response, especially for genes repressed in response to stress. This is interesting as an initial observation; I honestly don't know what to make of it, and the paper doesn't help me out too much. I don't mean to minimize the potential significance, but it's really unclear what this means, and the proteomics don't really inform the cell biology, nor vice versa. Just to take one possibility, maybe point (3) explains point (4): mitosis-delayed cells get very big, and maybe big cells are stressed. There must after all be some reason why cells do have a characteristic cell size, with evolved mechanisms to keep it there.

The idea is proposed that slowing cell growth at large cell sizes amounts to a cryptic or de facto size control. This will certainly work. In the limit, if cells stopped growing altogether after some hours of arrest, then whenever they divide thereafter, daughters will be about the same size.

The discussion makes the interesting point that early stages of mitotic slippage could work with mechanisms targetable in a therapeutic context. This is certainly something worth thinking about, and indeed has been a subject of a lot of discussion in the literature.

Overall, the experimental system used here is a nice one, and the results are carefully analyzed and fairly described. However, I really can't see too much that is really new or surprising here; the

proteomics could get there, but as noted, this seems somewhat preliminary in terms of biological context and specificity. The basic finding that slippage correlates with drops in cyclin B levels is cleanly demonstrated here, which is certainly useful, and consistent with a good deal of previous work in yeast and animals. On the other hand, the *cdh1* results show that slippage including mitotic exit and a new cell cycle does ultimately occur with no cyclin B degradation at all (Fig 2C). What (if anything) is inhibiting Clb-CDK activity in this context?

Minor points:

In some experiments, a Clb2 drop is used as a standin for anaphase - this is problematic since Clb2 degradation reflects CDC20 somewhat and CDH1 a lot (in fact the original literature said that CDH1 was the ONLY Clb2-degrading activity). So probably most of what is being detected as Clb2 is strongly degraded is CDH1-dependent, while CDH1 is entirely dispensable for anaphase. Similarly, equating 'prometaphase' with beginning of Clb2 accum is argumentative - prometaphase has cytological meaning not at all examined here.

The use of 'G1' to refer to 'generation 1' out of alpha-factor arrest conflicts with the standard use of G1 for gap phase 1, a usage also employed in this MS. It took me a while to figure out the 'generation 1' thing, which made the figures initially incomprehensible.

cdh1: missing control is Clb2 levels in *cdh1*-del without SAC activation

S4: *cdc20*-137 should be in caps because dominant.

Referee #2 Review

Report for Author:

When treated with anti-microtubule agents, eukaryotic cells arrest in prometahase due to the activation of spindle assembly checkpoint (SAC). An extended SAC-induced mitotic arrest can induce programmed cell death. This cellular behavior has become the basis for the use of anti-microtubule agents as cancer therapeutics. However, cells are unable to impose SAC-induced mitotic arrest for long periods; as a result some cells overcome the arrest (mitotic slippage) and proceed to the subsequent cell cycle and resume proliferation. Mitotic slippage is thought to be one of the mechanisms for the development of resistance to anti-microtubule therapeutics. While some cells may inactivate the checkpoint to escape mitotic arrest, others enter the next cycle with activated checkpoint. Whether cells proliferating with activated checkpoint have altered cellular responses or cell cycle parameters is not clearly understood.

Since cell cycle controls and SAC regulation are largely conserved in yeast and human cells, Corno et al address this specific issue in budding yeast *S. cerevisiae*. They use two different ways to activate the checkpoint (GAL-MAD2 expression and *tub2*-401 mutation) and not the standard method of treatment with anti-microtubule agents to activate the SAC. The authors draw three major conclusions from this study: (i) cells that escape mitotic arrest remain SAC responsive in the subsequent cycles (ii) survival/proliferation of cells with active checkpoint requires Cdh1, a protein that is non-essential in normal cells (iii) Changes in protein expression in these cells correlates with the environmental stress response.

It is a detailed investigation into the division cycle of cells proliferating with active SAC. The

observation that a gene which is nonessential in normal cells becomes 'essential' in these cells is perhaps the most interesting finding of this study and may have therapeutic implication if it is extrapolated to human cells, at least in the context of antimetabolic drugs. The experiments are fairly well-controlled and generally support the conclusions (in a sort-of broad sense) that the authors have drawn from them. However there are a few caveats that need to be addressed:

Major points:

1. Although GAL-MAD2 can cause cells to arrest in prometaphase, these cells are not physiologically equivalent to the cells arrested in response to the disruption of kinetochore-microtubule connections (or loss of tension). For instance, in GAL-MAD2 cells Mps1 is not activated, the dynamics at the kinetochore is not the same as it would be in cells arrested in response to the disruption of kinetochore-microtubule connections etc. Same is true of the tub2-401 mutant (as mentioned below). These caveats should at least be discussed in the 'Discussion section' to put the derived-conclusion in context.
2. The two methods used in this study to activate checkpoint, i.e. GAL-MAD2 or tub2-401 mutation, seem to have large difference in their effect on the cell cycle with 'cell cycle length' and 'mitotic length' much longer in tub2-401 (Fig 1D). Does this reflect the difference in the extent of SAC activation by the two methods or it is due to additional cellular defects/delays (independent of the checkpoint activation) caused by the tub2-401 mutation? If the difference in timing is also contributed by additional defects caused by the tub2-401 mutation, drawing conclusions only in terms of SAC is not really justified.
3. From the data, it is not clear what proportion of GAL-MAD2 and tub2-401 escape mitosis to continue on to the next cycle.
4. Fig 1D and Fig S1E show the time from 'Clb2-increase to Clb2-decrease' and 'maximum level of Clb2', respectively. It will be informative if the author has also shown the level of Clb2 at the time of exit from the first cycle in both GAL-Clb2 and tub2-401 strains. The level of Clb2 at exit would affect the progression through the next cycle (this relates to the point 5 below) and is linked to the heightened requirement for Cdh1 in the subsequent cycle.
5. Fig 2 relates to one of the main conclusions of this study that Cdh1 becomes 'essential' in cells proliferating with active checkpoint. Qualitatively, *cdh1Δ* affects GAL-MAD2 cells more than tub2-401 cells (at 20C) (Fig 2A) even though tub2-401 cells show much greater delay in cell cycle length and mitotic length? What is the explanation for this observation? Is it because the 'strength of the checkpoint' is relatively greater in GAL-MAD2? If so, then it will suggest that the longer delays in cell cycle/mitotic timings observed in tub2-401 cells are not only due to active checkpoint but other cell cycle defects.
6. Why are the plots similar to ones shown for GAL-CLB2 and GAL-CLB2 *cdh1Δ* shown for tub2-401 (Fig 2C)?
7. There is no evidence (or very scant) provided that Cdc20 is inhibited in the subsequent cycles (and therefore cells are more heavily dependent on Cdh1)?
8. Why is the shape of GAL-MAD2 cells different from that of tub2-401 (Fig S3D)? The shape of GAL-MAD2 cells is reminiscent of cells that fail to switch from polarized growth to isotropic growth (or cells lacking Clb2). Does overexpression of Mad2 affect cellular processes other than the activation of the SAC?
9. As shown in Fig 3C, GAL-MAD2 cell size has much greater scatter compared to tub2-401. This is also apparent in Fig S3B (local growth rate vs Clb2 level). Given this, is the general conclusion that SAC active cells reach growth rate saturation really justified?
10. The authors conclude that larger cells do not have a size control at anaphase entry. The argument was that larger cells will 'dilute the checkpoint effectors'. Given that *S. cerevisiae* has closed mitosis (i.e. nuclear membrane remains intact during mitosis) and that SAC machinery and its action are all within the nucleus, it is the nuclear size that is relevant to the 'dilution argument' and not the overall size of the cells. It is not surprising that cell size does not show any correlation to the

timing of anaphase entry.

11. The authors emphasize that cells proliferating with active checkpoint remain checkpoint responsive in the subsequent cycles. However, in both systems they have used, namely GAL-MAD2 and tub2-401 mutation, MAD2 is expressed and tub2-401 mutation are present in the subsequent cycles. Hence, it is not surprising that cells respond to these perturbations in every cycle and exhibit the observed delays. Hence the point about 'checkpoint responsiveness' needs to be put in proper perspective

Minor concerns

There are a few grammatical errors and some awkward phrasing in the text. These should be amended.

Referee #3 Review

Report for Author:

In their manuscript 'Cellular response upon proliferation in the presence of an active mitotic checkpoint' Corno et al. study proliferation of budding yeast cells despite an active spindle assembly checkpoint. They present live-cell microscopy and mass spectrometry data on two different systems that allow permanent activation of the SAC. The presented work is scientifically sound and the data are interesting, and I therefore think that the manuscript is in principle suited for publication in this journal. My major concerns that need to be addressed before publication regard some of the conclusions. In addition, while I do not disagree with the results obtained from the modeling, as explained below, I suspect that the authors could have taken the modeling approach a little more seriously and test their conclusions in a quantitative manner.

- In Fig 3 B and C, the authors present single cell data showing size at anaphase as a function of size at prometaphase. Cells with an active SAC grow significantly more before entering anaphase and the dependence of size at anaphase on size at PM becomes much weaker. Based on these data, the authors draw the conclusion that there is no size control mechanism. Based on the data, I would have concluded the opposite. Maybe this disagreement boils down to semantics. What does size control mean? If the authors understand by size control that there is one precise and deterministic size at which cells enter anaphase, then there is obviously no size control. It is however important to note that also in the 'traditional G1/S' size control, this is not the case. Instead, what the field calls size control in this case is the phenomenon that the duration of (and thus the relative growth during) pre-Start G1 decreases with birth size. However, this effect is highly stochastic and only on average we see that smaller born cells partially compensate for the small initial size by longer growth before entering S phase. Especially for the tub2-401 data this seems to be similarly the case. I therefore suggest that the authors test for the presence of size control by plotting more directly: 1) how does the duration of PM-A depend on cell size at PM? 2.) Is it possible to extract a rate of entering anaphase as a function of cell size at a given point, as described by Chandler-Brown et al., Curr Biol 2017 for G1/S control? If this rate depends on cell size, I would interpret this (by definition) as 'size control'.

- In Fig. 3C and D the authors show data on the size dependent growth rate and conclude that large cells in the presence of an active SAC slow down growth and show a 'saturation of area'. Also in the modelling part the authors write that the growth rate decreases. Based on the data, I disagree with these conclusions. In Fig. 3C, cells clearly keep growing to the very end, and consistent with this observation Fig. 3D shows that at all sizes the local growth rate remains

positive. For the Gal1-Mad2 I don't even see a decrease of the growth rate. I suspect that part of the confusion originates from somewhat sloppy wording in the manuscript. Do the authors conclude that size saturates (which I don't see in the data) or do they conclude that the growth rate saturates, i.e. that area keeps increasing linearly with size (which I would agree with)? Is it the size vs time relation or the size dependent growth rate that is compared to a logistic curve? I think this part of the manuscript would strongly benefit from more precise writing, and it might also be interesting for the authors to consider that also WT yeast cells show a transition from exponential to linear volume growth at large volumes (e.g. Chandler-Brown et al., but also others).

- I am not an expert on mass spectrometry, but I am not sure if I should agree with the final conclusion that 'large parts of the proteome are commonly regulated' in cells with an active SAC, given the weak correlation of $r=0.17$ shown in Fig. 4B. To be fair, the authors provide the relevant statistics so that the reader can let the data speak for themselves, but I feel that the conclusions are slightly overstated.
 - In Fig. 5 the authors show a toy model that explains how a size-dependent growth rate together with a stochastic slow transition through the cell cycle results in a steady state size distribution. While not wrong, I feel like the authors give away the opportunity to build a more quantitative but not necessarily more complex model based on their own data. Why not base both the size-dependent growth rate and cell cycle duration distributions directly on the data provided in Fig. 1-3? I would have expected a comparison of model assumption and experimental data for these relations in Fig. 5. If for example the logistic function does not accurately follow the data in Fig. 3C, an alternative approach would be to fit the data with an appropriately complex function and use the obtained relation for the simulations. Once the model is based on the actual data, it should be possible to use the model to test if the assumptions recapitulate independent observations. For example, should the model not quantitatively reproduce the 'size control' data in Fig. 3B, which strongly support the claim that the size-dependent growth rate is the origin of the 'passive size control' preventing diverging cell size.
-

March 13, 2019

Re: Life Science Alliance manuscript #LSA-2019-00380-T

Andrea Ciliberto
IFOM, Fondazione Istituto FIRC di Oncologia Molecolare
Via Adamello 16
Milan 20139
Italy

Dear Dr. Ciliberto,

Thank you for transferring your manuscript entitled "Cellular response upon proliferation in the presence of an active mitotic checkpoint" to Life Science Alliance. The manuscript was assessed by expert reviewers at another journal before, and the editors transferred those reports to us with your permission.

The reviewers appreciate the technical quality of your work but found the overall conceptual advance somewhat limited. This is not a concern for publication in Life Science Alliance and we would thus like to invite you to submit a revised version for publication here. We would expect a point-by-point response to the concerns raised and accordingly inclusion of the requested controls as well as changes to data representation/interpretation and discussion.

Thank you for this interesting contribution to Life Science Alliance. We are looking forward to receiving your revised manuscript.

Sincerely,

B. MANUSCRIPT ORGANIZATION AND FORMATTING:

Referee #1:

The paper concerns the phenomenon of mitotic 'slippage', where cells with an activated spindle assembly checkpoint (SAC) nevertheless proceed into mitosis. Two means of activating the SAC were employed: 'gratuitous' induction by overexpressing Mad2 in the absence of significant spindle perturbation; and partial spindle dysfunction with the tub2-401 cs mutation at semipermissive temperature. In general, results are considered relevant to slippage if found in both systems. This is a good approach that bypasses the substantial difference between the approaches. There could still be an issue that neither approach is necessarily equivalent to complete checkpoint activation; tub2-401 cells are viable under these conditions so most spindles are fully functional most of the time (so at least some of the cycles observed are in fact normal cycles - thus no slippage required!). In contrast, Mad2 overexpression does nothing much to the spindle (though it is not inert, as shown here), so if a given level of Mad2 fully activates the checkpoint, that makes for easy interpretation. But do we really know this? After all, the normal checkpoint DOESN'T work by increasing Mad2 expression, rather by converting protein-protein interactions of Mad2 and the host of other kinetochore/MCC proteins. However, as noted, the use of both approaches does deal with these issues reasonably well.

We thank the reviewer for appreciating our experimental approach.

What is learned? (1) Clb2 (mitotic cyclin) rises to high levels during the checkpoint arrests, and declines sharply during slippage exits. This is consistent with previous work in various systems. It is shown here that this pattern repeats in subsequent cycles, which is interesting to know (while the well-known necessity for B-cyclin-CDK activity to drop to allow mitotic exit makes it a reasonably likely result going in, it was possible that in later cycles cells might somehow 'discover' some other means, such as SIC1 overexpression). This finding indicates that slipping cells are not stably 'adapted' to a continued checkpoint signal.

We agree with the reviewer, the possibility that cells would become refractory to the stimulus (as in the case of α -factor induced G1) was a possibility worth testing. This was in fact our starting biological question, as we discuss also in the last answer to reviewer 2.

(2) Slippage is at least somewhat dependent on CDH1, an alternate APC activator to CDC20 (the target of SAC inhibition). This is interesting; essentiality of CDH1 in cases where CDC20 is compromised has been reported in various contexts (including in mitotic slippage in animal cells). Ref. 9 (from the authors' laboratory) showed no requirement for CDH1 for the first anaphase in this same system with GAL-MAD2 (at least, for loss of a scored mitotic spindle). What is shown here is that subsequent mitotic exit is considerably slower without CDH1. I guess this suggests incomplete CDC20 activation, enough to get at securin but not enough to clear CLB2 low enough to allow mitotic exit without assistance from CDH1 (this is my own interpretation because I couldn't exactly get the authors' ideas here). This should all be explained more clearly because as I understand it, this is a very different perspective on the role of CDH1 in slippage from that proposed in ref. 9 (where the simple statement was that CDH1 was not involved).

We thank the reviewer for helping us clarifying this point. What the reviewer suggests is also our interpretation, which we have produced originally in Bonaiuti et al, Curr Biol, 2018 and which we state more clearly in the revised Discussion of the current manuscript. However, we disagree with the fact that this result differs from what we reported in our previous work. In particular, Figure S3A shows the same result as Figure 1E in Vernieri et al, JCB 2013, namely that the timing of

anaphase entry (here, nuclear segregation) in the first slippage event is not affected by the deletion of *CDH1*. What we report here (which was not investigated in the previous publication) is that the following mitotic exit and G1 are instead affected by the lack of *CDH1*. This is now more explicitly stated in the Discussion.

(3) There is some examination of cell size under these conditions. The checkpoint-delayed cells are larger than unperturbed cells, which is completely expected (since cell growth continues for some time at cell cycle blocks, as first shown by Hartwell in the 1970's). It is further shown that cell growth rate slows as cells undergo prolonged checkpoint delays. This is potentially interesting but needs to be compared to growth rates of similar-sized cells due to different blocks, in order to disentangle a checkpoint-specific response from truly huge cells slowing down because they only have one nucleus, only have one (or two) genome equivalents, have a low surface-to-volume ratio, etc etc.

The reviewer's comment is interesting, and is in line with what is reported in a recent manuscript, which we had not cited since it came out while we were under revision. Here, the Amon lab (Neurohr et al, Cell, 2019) shows that large cells decrease growth because they are large, due to the lowered nucleo-cytoplasmic ratio. These results are extremely interesting, and we now mention them, but we believe that the phenotypes we observe, among which slower growth rate in SAC-active cells, are not due to the large size *per se*. SAC-active cells are between 2-3 times larger than unperturbed cells (Figure S4A), with the largest cells being below 200 fL in *tub2-401*, and twice as much in *GALI-MAD2*. In the Neurohr paper, Figure 3B, it is reported that growth does not saturate within this volume range but from 500 fL upward. In fact, our cells are closer to the negative controls of the Neurohr paper (CHX, low glucose) than to large cells. Thus, we believe that the slowing down is not due simply to the larger size, but to the prolongation of mitosis induced by the chronic activation of the mitotic checkpoint. We discuss this point in the new version of the paper, in the Discussion.

(4) The SAC-active cells induce a proteomic response with overlap to a general stress response, especially for genes repressed in response to stress. This is interesting as an initial observation; I honestly don't know what to make of it, and the paper doesn't help me out too much. I don't mean to minimize the potential significance, but it's really unclear what this means, and the proteomics don't really inform the cell biology, nor vice versa. Just to take one possibility, maybe point (3) explains point (4): mitosis-delayed cells get very big, and maybe big cells are stressed. There must after all be some reason why cells do have a characteristic cell size, with evolved mechanisms to keep it there.

As mentioned above, we think we can exclude that what we observe is simply the effect of cells reaching a very large size since our cells do not reach the large size obtained in the Neurohr paper. Instead, we find it interesting that 20 minutes of treatment with benomyl (hardly enough time to make cells too big) generates a response that in terms of global protein expression clusters with ours (Figure 4C). In our view, it is specifically the mitotic arrest that generates a condition of stress, slows down growth and generates the synthetic lethality with *cdh1*Δ. In other words, we believe that these cells are not simply big, but set out a series of responses to cope with the ectopic mitotic delay.

The idea is proposed that slowing cell growth at large cell sizes amounts to a cryptic or de facto size control. This will certainly work. In the limit, if cells stopped growing altogether after some hours of arrest, then whenever they divide thereafter, daughters will be about the same size.

We thank the reviewer for appreciating our hypothesis.

The discussion makes the interesting point that early stages of mitotic slippage could work with mechanisms targetable in a therapeutic context. This is certainly something worth thinking about, and indeed has been a subject of a lot of discussion in the literature.

In the new version of the manuscript, we included additional experiments and simulations that prove this point even more effectively. We show that when the checkpoint stimulus is lifted, cells go back to their unperturbed state in approximately three hours (Figure 5). This reversibility of the phenotype that we observe gives additional weight to the idea of targeting specifically SAC-active cells before they become resistant.

*Overall, the experimental system used here is a nice one, and the results are carefully analyzed and fairly described. However, I really can't see too much that is really new or surprising here; the proteomics could get there, but as noted, this seems somewhat preliminary in terms of biological context and specificity. The basic finding that slippage correlates with drops in cyclin B levels is cleanly demonstrated here, which is certainly useful, and consistent with a good deal of previous work in yeast and animals. On the other hand, the *cdh1* results show that slippage including mitotic exit and a new cell cycle does ultimately occur with no cyclin B degradation at all (Fig 2C). What (if anything) is inhibiting Clb-CDK activity in this context?*

This is surely an interesting question, we agree with the reviewer. The first hypothesis is obviously that counteracting phosphatases (Cdc14 in budding yeast) may play a fundamental role. The fact that mitotic exit requires a careful balance of kinases and phosphatase activities when cells cannot fully degrade Clb2 has been clearly stated in the past, in quantitative terms for example by the Cross lab (Drapkin, MSB, 2009). We now discuss this point in the Discussion.

Minor points:

In some experiments, a Clb2 drop is used as a standin for anaphase - this is problematic since Clb2 degradation reflects CDC20 somewhat and CDH1 a lot (in fact the original literature said that CDH1 was the ONLY Clb2-degrading activity). So probably most of what is being detected as Clb2 is strongly degraded is CDH1-dependent, while CDH1 is entirely dispensable for anaphase. Similarly, equating 'prometaphase' with beginning of Clb2 accum is argumentative - prometaphase has cytological meaning not at all examined here.

We thank the reviewer for pointing this out. We agree that Clb2 synthesis and degradation may have shortcomings for detecting mitotic entry and anaphase. Yet, we think they are reasonable first approximations. The fast Cdh1-dependent degradation of Clb2, which we take for anaphase onset, needs to be triggered by Cdc20. Hence, the initial point of Clb2 degradation is a good readout for APC/C^{Cdc20} activation, which also triggers separation of sister chromatids and thus anaphase. Likewise, the accumulation of the essential mitotic cyclin is a good readout of entry into mitosis. Our choice is quite handy for additional reasons: (i) it allows us to use one marker only; (ii) timing of mitotic entry and anaphase can be estimated quantitatively via image analysis; and (iii) Clb2 is a direct target of APC/C^{Cdc20}, which is the target of the mitotic checkpoint. To address the point raised by the reviewer, we emphasize in the text that we use parameters that allow the *indirect* identification of mitotic entry and anaphase onset.

The use of 'G1' to refer to 'generation 1' out of alpha-factor arrest conflicts with the standard use of G1 for gap phase 1, a usage also employed in this MS. It took me a while to figure out the 'generation 1' thing, which made the figures initially incomprehensible.

We thank the reviewer for pointing this out. We understand that our terminology was a source of confusion. We now introduce explicitly the concept of generations in Figure S2A.

cdh1: missing control is Clb2 levels in cdh1-del without SAC activation.

We have introduced this control, which confirms how also in cycling cells Cdh1 is essential for proper Clb2 degradation (Figure S3B). We find that while *GAL1-MAD2 cdh1Δ* and *cdh1Δ* cells are both defective in Clb2 degradation, the increase of Clb2 levels at anaphase is quite different. In SAC-active *cdh1Δ* cells (Figure 2C), we find 2.8 times the value measured in wild type cells, while in *cdh1Δ* alone they are 1.4 times higher (Figure S3B). As reported by Drapkin et al. MSB 2009, stable Clb2 above 2X endogenous levels introduces dose-dependent delays in mitotic exit and defects in proliferation. Accordingly, proliferation defects are much more pronounced in *GAL1-MAD2 cdh1Δ* (Figure 2A).

S4: cdc20-137 should be in caps because dominant.

Changed as suggested.

Referee #2:

When treated with anti-microtubule agents, eukaryotic cells arrest in prometaphase due to the activation of spindle assembly checkpoint (SAC). An extended SAC-induced mitotic arrest can induce programmed cell death. This cellular behavior has become the basis for the use of anti-microtubule agents as cancer therapeutics. However, cells are unable to impose SAC-induced mitotic arrest for long periods; as a result some cells overcome the arrest (mitotic slippage) and proceed to the subsequent cell cycle and resume proliferation. Mitotic slippage is thought to be one of the mechanisms for the development of resistance to anti-microtubule therapeutics. While some cells may inactivate the checkpoint to escape mitotic arrest, others enter the next cycle with activated checkpoint. Whether cells proliferating with activated checkpoint have altered cellular responses or cell cycle parameters is not clearly understood.

*Since cell cycle controls and SAC regulation are largely conserved in yeast and human cells, Corno et al address this specific issue in budding yeast *S. cerevisiae*. They use two different ways to activate the checkpoint (*GAL-MAD2* expression and *tub2-401* mutation) and not the standard method of treatment with anti-microtubule agents to activate the SAC. The authors draw three major conclusions from this study: (i) cells that escape mitotic arrest remain SAC responsive in the subsequent cycles (ii) survival/proliferation of cells with active checkpoint requires Cdh1, a protein that is non-essential in normal cells (iii) Changes in protein expression in these cells correlates with the environmental stress response.*

It is a detailed investigation into the division cycle of cells proliferating with active SAC. The observation that a gene which is nonessential in normal cells becomes 'essential' in these cells is perhaps the most interesting finding of this study and may have therapeutic implication if it is extrapolated to human cells, at least in the context of antimitotic drugs. The experiments are fairly

well-controlled and generally support the conclusions (in a-sort-of broad sense) that the authors have drawn from them. However there a few caveats that need to be addressed:

Major points:

1. Although GAL-MAD2 can cause cells to arrest in prometaphase, these cells are not physiologically equivalent to the cells arrested in response to the disruption of kinetochore-microtubule connections (or loss of tension). For instance, in GAL-MAD2 cells Mps1 is not activated, the dynamics at the kinetochore is not the same as it would be in cells arrested in response to the disruption of kinetochore-microtubule connections etc. Same is true of the tub2-401 mutant (as mentioned below). These caveats should at least be discussed in the 'Discussion section' to put the derived-conclusion in context.

We understand the point of the reviewer, and that is why we used both experimental systems to address our biological question. To address the reviewer's concerns about the possible caveats coming with the *tub2-401*, we performed a new experiment (Figure S2C). We followed with live-cell imaging the cell cycle duration in *tub2-401* with or without the essential checkpoint component *MAD2*. The new results show that *tub2-401 mad2Δ* cells go through mitosis very much like wild type cells. This result, which complements the genetic analysis in Figure S1B, shows that the mitotic checkpoint is responsible for the delay observed in *tub2-401* cells. The same argument applies to *GAL1-MAD2* cells, since *GAL1-MAD2 mad3Δ* and wild type cells grow similarly – as we reported in Figure 2A in Mariani, Chiroli et al. Curr. Biol. 2012.

2. The two method used in this study to activate checkpoint, i.e. GAL-MAD2 or tub2-401 mutation, seem to have large difference in their effect on the cell cycle with 'cell cycle length' and 'mitotic length' much longer in tub2-401 (Fig 1D). Does this reflect the difference in the extent of SAC activation by the two methods or it is due to additional cellular defects/delays (independent of the checkpoint activation) caused by the tub2-401 mutation? If the difference in timing is also contributed by additional defects caused by the tub2-401 mutation, drawing conclusions only in terms of SAC is not really justified.

Despite the different mechanisms of action that trigger the mitotic checkpoint in *GAL1-MAD2* and *tub2-401* cells, we believe that the main difference here stems from the different temperatures at which we performed the two experiments. While *GAL1-MAD2* cells grew at 30° C, *tub2-401* cells are grown at lower temperature, and this of course has a large influence on the absolute duration of mitosis and of the cell cycle. Instead, we do not think the phenotype of the *tub2-401* mutant reflects additional effects of this mutant: for the reasons explained in the above paragraph, and because the properties of SAC-active cells that we emphasize are those shared with the *GAL1-MAD2* system. In the text, we now mention the different temperatures as a rationale for the quantitative difference between the two results (i.e., longer delay observed in *tub2-401*).

3. From the data, it is not clear what proportion of GAL-MAD2 and tub2-401 escape mitosis to continue on to the next cycle.

This is an important point, for which we thank the reviewer. Roughly 70% of cells manage to escape from the arrest and actively proliferate. This is now shown in Figure S2B.

4. Fig 1D and Fig S1E show the time from 'Clb2-increase to Clb2-decrease' and 'maximum level of Clb2', respectively. It will be informative if the author has also shown the level of Clb2 at the time

of exit from the first cycle in both GAL-Clb2 and tub2-401 strains. The level of Clb2 at exit would affect the progression through the next cycle (this relates to the point 5 below) and is linked to the heightened requirement for Cdh1 in the subsequent cycle.

This is another interesting point, for which we thank the reviewer. We have performed the analysis suggested (Figure S2E). These values are slightly larger than in unperturbed cells, and provide the basis for the argument put forward by the reviewer.

5. Fig 2 relates to one of the main conclusion of this study that Cdh1 becomes 'essential' in cells proliferating with active checkpoint. Qualitatively, cdh1Δ affects GAL-MAD2 cells more than tub2-401 cells (at 20C) (Fig 2A) even though tub2-401 cells show much great delay in cell cycle length and mitotic length? What is the explanation for this observation? Is it because the 'strength of the checkpoint' is relatively greater GAL-MAD2? If so, then it will suggest that the longer delays in cell cycle/mitotic timings observed in tub2-401 cells are not only due to active checkpoint but other cell cycle defects

Given the difference in temperature, we believe that it is hard to conclude that *tub2-401* is more heavily affected by the deletion of *CDH1*. Instead, as we argue above, we believe that one can conclude quite safely that in both experimental system there is a synthetic effect of *cdh1Δ* and checkpoint activation.

6. Why the plots similar to ones shown for GAL-CLB2 and GAL-CLB2 cdh1Δ shown for tub2-401 (Fig2C)?

The observation of the reviewer is correct. The reason is primarily technical. Taking movies at the restrictive temperature is a very laborious and error-prone process. For this reason, we decided to focus our efforts on the *tub2-401* experiments (Figure 1 and related supplementary figures). After a few unlucky attempts with *tub2-401 cdh1Δ* we gave up, also because we felt that the genetic analysis in Figure 2A-B shows the same synthetic effect in the two experimental systems.

7. There is no evidence (or very scant) provided that Cdc20 is inhibited in the subsequent cycles (and therefore cells are more heavily dependent on Cdh1)?

We disagree with the reviewer on this point. The evidence we present is quite compelling. Clb2 is a key substrate of APC/C^{Cdc20} at the metaphase-to-anaphase transition. Cdh1 drives Clb2 degradation after this point, but cannot replace Cdc20 in driving anaphase onset, as cells deprived of Cdc20 are permanently arrested before anaphase. Thus, the fact that Clb2 is similarly stabilized and transition to anaphase equally delayed during the early and late cycles strongly suggests that Cdc20 is constantly inhibited. If Cdc20 would not be inhibited, cells would undergo faster cell divisions in the later cycles, something that we do not observe (Figure S2F-G).

8. Why is the shape of GAL-MAD2 cells different from that of tub2-401 (Fig S3D)? The shape of GAL-MAD2 cells is reminiscent of cells that fail to switch from polarized growth to isotropic growth (or cells lacking Clb2). Does overexpression of Mad2 affect cellular processes other than the activation of the SAC?

This is an interesting question. We are not sure about the answer. Growth in cells overexpressing Mad2 seems to be more polarized than in *tub2-401*. Based on work from Lew and Reed, one would

conclude that this is due to Clns-related Cdk1 activity. Why that should be the case in cells overexpressing Mad2 is not obvious. Regardless of these morphological differences, it is very interesting that in quantitative terms the growth curves are very similar. When we plot the local growth rate -- Figure S5A -- we can see that in both cases the rates reach their maximum at around $60 \mu\text{m}^2$. Thus, although we cannot provide an answer, we think that the morphological differences do not affect the more substantial similarities revealed in our analysis.

9. As shown in Fig 3C, GAL-MAD2 cell size has much greater scatter compared tub2-401. This is also apparent in Fig S3B (local growth rate vs Clb2 level). Given this, is the general conclusion that SAC active cells reach growth rate saturation really justified?

We thank the reviewer for raising this point, which is also shared by reviewer 3. We agree that an exponential growth with a linear asymptote is a better description of our data. Hence, we replaced the logistic curve with such a growth rate (see Figure 3D). The results we presented have not changed noticeably. For a longer discussion see our response to reviewer 3.

10. The authors conclude that larger cells do not have a size control at anaphase entry. The argument was that larger cells will 'dilute the checkpoint effectors'. Given that S. cerevisiae has closed mitosis (i.e. nuclear membrane remains intact during mitosis) and that SAC machinery and its action are all within the nucleus, it is the nuclear size that is relevant to the 'dilution argument' and not the overall size of the cells. It is not surprising that cell size does not show any correlation to the timing of anaphase entry.

We agree with the reviewer, we removed this argument from the paper.

11. The authors emphasize that cells proliferating with active checkpoint remain checkpoint responsive in the subsequent cycles. However, in both systems they have used, namely GAL-MAD2 and tub2-401 mutation, MAD2 is expressed and tub2-401 mutation are present in the subsequent cycles. Hence, it is not surprising that cells respond to these perturbations in every cycle and exhibit the observed delays. Hence the point about 'checkpoint responsiveness' needs to be put in proper perspective.

This was actually our question from the very beginning: whether cells that are continuously under checkpoint activating conditions eventually become insensitive, ie manage to ignore the stimulus. This is after all what has been observed in cells arrested in G1 by α -factor treatment (from the Barral lab – Caudron et al. Cell 2013). So, we believe that one major conclusion is that actually SAC-active cells do not become refractory to the stimulus. Instead, they remain committed to the checkpoint even after having divided with an active checkpoint. We make this point clearly in the revised Discussion.

Minor concerns

There are a few grammatical errors and some awkward phrasing in the text. These should be amended.

We have changed the text where we thought it needed to be amended.

Referee #3:

In their manuscript 'Cellular response upon proliferation in the presence of an active mitotic checkpoint' Corno et al. study proliferation of budding yeast cells despite an active spindle assembly checkpoint. They present live-cell microscopy and mass spectrometry data on two different systems that allow permanent activation of the SAC. The presented work is scientifically sound and the data are interesting, and I therefore think that the manuscript is in principle suited for publication in this journal. My major concerns that need to be addressed before publication regard some of the conclusions. In addition, while I do not disagree with the results obtained from the modeling, as explained below, I suspect that the authors could have taken the modeling approach a little more seriously and test their conclusions in a quantitative manner.

• In Fig 3 B and C, the authors present single cell data showing size at anaphase as a function of size at prometaphase. Cells with an active SAC grow significantly more before entering anaphase and the dependence of size at anaphase on size at PM becomes much weaker. Based on these data, the authors draw the conclusion that there is no size control mechanism. Based on the data, I would have concluded the opposite. Maybe this disagreement boils down to semantics. What does size control mean? If the authors understand by size control that there is one precise and deterministic size at which cells enter anaphase, then there is obviously no size control. It is however important to note that also in the 'traditional G1/S' size control, this is not the case. Instead, what the field calls size control in this case is the phenomenon that the duration of (and thus the relative growth during) pre-Start G1 decreases with birth size. However, this effect is highly stochastic and only on average we see that smaller born cells partially compensate for the small initial size by longer growth before entering S phase. Especially for the tub2-401 data this seems to be similarly the case. I therefore suggest that the authors test for the presence of size control by plotting more directly: 1) how does the duration of PM-A depend on cell size at PM? 2.) Is it possible to extract a rate of entering anaphase as a function of cell size at a given point, as described by Chandler-Brown et al., Curr Biol 2017 for G1/S control? If this rate depends on cell size, I would interpret this (by definition) as 'size control'.

We thank the reviewer for raising this point. We agree with the reviewer's suggestion, and we now changed our interpretation, and argue that indeed there is a mechanism that compensates for size differences. We could not conclude this from the analyses suggested by the reviewer, since analysis (1), plotting duration vs size at PM, did not provide a positive answer (ie, we did not see a correlation between size at PM and the duration of mitosis -- Figure S4C); and analysis (2) could not be performed due to the paucity of cells. However, when we plotted the correlation between $\ln(\text{Area at PM}/\text{Area at Ana})$ and $\ln(\text{Area at PM})$, we did find a negative correlation (Figure 3B), which is informative of a size control (according to Cadart et al. Nat. Commun. 2018). Thus, we now confirm the presence of a mechanism that compensates for differences in sizes at PM.

• In Fig. 3C and D the authors show data on the size dependent growth rate and conclude that large cells in the presence of an active SAC slow down growth and show a 'saturation of area'. Also in the modelling part the authors write that the growth rate decreases. Based on the data, I disagree with these conclusions. In Fig. 3C, cells clearly keep growing to the very end, and consistent with this observation Fig. 3D shows that at all sizes the local growth rate remains positive. For the Gal1-Mad2 I don't even see a decrease of the growth rate. I suspect that part of the confusion originates from somewhat sloppy wording in the manuscript. Do the authors conclude that size saturates (which I don't see in the data) or do they conclude that the growth rate saturates, i.e. that area keeps increasing linearly with size (which I would agree with)? Is it the size vs time relation or the size dependent growth rate that is compared to a logistic curve? I think this part of the manuscript would strongly benefit from more precise writing, and it might also be interesting for

the authors to consider that also WT yeast cells show a transition from exponential to linear volume growth at large volumes (e.g. Chandler-Brown et al., but also others).

We agree with the reviewer, and we revised our terminology. We say that growth rate saturates, and accordingly, we do not use the logistic to reproduce our data, but rather an exponential that becomes linear when size increases, as suggested by the reviewer. Along this line, we also mention the manuscript from Chandler-Brown as suggested. Hence, in summary, we fully adopt the point suggested by the reviewer, and we now say that rate keeps increasing linearly with size (see Figure 3D).

• I am not an expert on mass spectrometry, but I am not sure if I should agree with the final conclusion that 'large parts of the proteome are commonly regulated' in cells with an active SAC, given the weak correlation of $r=0.17$ shown in Fig. 4B. To be fair, the authors provide the relevant statistics so that the reader can let the data speak for themselves, but I feel that the conclusions are slightly overstated.

We understand the reviewer's point. We toned down our claim, saying that 'a sizeable fraction of the proteome is commonly regulated'.

• In Fig. 5 the authors show a toy model that explains how a size-dependent growth rate together with a stochastic slow transition through the cell cycle results in a steady state size distribution. While not wrong, I feel like the authors give away the opportunity to build a more quantitative but not necessarily more complex model based on their own data. Why not base both the size-dependent growth rate and cell cycle duration distributions directly on the data provided in Fig. 1-3? I would have expected a comparison of model assumption and experimental data for these relations in Fig. 5. If for example the logistic function does not accurately follow the data in Fig. 3C, an alternative approach would be to fit the data with an appropriately complex function and use the obtained relation for the simulations. Once the model is based on the actual data, it should be possible to use the model to test if the assumptions recapitulate independent observations. For example, should the model not quantitatively reproduce the 'size control' data in Fig. 3B, which strongly support the claim that the size-dependent growth rate is the origin of the 'passive size control' preventing diverging cell size.

We changed the model, as mentioned above. Following the reviewer's suggestion, it resembles more closely the experimental data. Yet, we believe that we cannot perform the analysis proposed by the reviewer because that would require a much more complex model. What we have now is a model that does not include Clb2 levels: we only have cycle time and growth. If we were to introduce different cell cycle phases, as would be needed to reproduce the suggested correlations (PM and A are identified by increasing and decreasing Clb2 levels), we would need to produce a much more complicated model, with more parameters and more unknowns. Instead, our idea was to have a simple model to put forward the idea nicely described by the reviewer that "the size-dependent growth rate is the origin of the 'passive size control' that prevents diverging cell size".

April 25, 2019

RE: Life Science Alliance Manuscript #LSA-2019-00380-TR

Dr. Andrea Ciliberto
IFOM, Fondazione Istituto FIRC di Oncologia Molecolare
Via Adamello 16
Milan 20139
Italy

Dear Dr. Ciliberto,

Thank you for submitting your revised manuscript entitled "Cellular response upon proliferation in the presence of an active mitotic checkpoint". I appreciate the introduced changes and additional data and would be happy to publish your paper in Life Science Alliance pending final revisions necessary to meet our formatting guidelines:

- please make a supplementary figure out of current figure SM1 (in methods section), this will allow easy in-line display in the HTML version of the paper
- please remove the superfluous 'for' in Figure 6 "Cdh1 is required..."
- please provide all tables as word doc files
- please link your profile in our submission system to your ORCID iD, you should have received an email with instructions on how to do so

A. FINAL FILES:

-- Summary blurb (enter in submission system): A short text summarizing in a single sentence the study (max. 200 characters including spaces). This text is used in conjunction with the titles of

papers, hence should be informative and complementary to the title. It should describe the context and significance of the findings for a general readership; it should be written in the present tense and refer to the work in the third person. Author names should not be mentioned.

B. MANUSCRIPT ORGANIZATION AND FORMATTING:

Sincerely,

Andrea Leibfried, PhD
Executive Editor
Life Science Alliance
Meyershofstr. 1
69117 Heidelberg, Germany
t +49 6221 8891 502
e a.leibfried@life-science-alliance.org
www.life-science-alliance.org

April 29, 2019

RE: Life Science Alliance Manuscript #LSA-2019-00380-TRR

Dr. Andrea Ciliberto
IFOM, Fondazione Istituto FIRC di Oncologia Molecolare
Via Adamello 16
Milan 20139
Italy

Dear Dr. Ciliberto,

Thank you for submitting your Research Article entitled "Cellular response upon proliferation in the presence of an active mitotic checkpoint". It is a pleasure to let you know that your manuscript is now accepted for publication in Life Science Alliance. Congratulations on this interesting work.

DISTRIBUTION OF MATERIALS:

Again, congratulations on a very nice paper. I hope you found the review process to be constructive and are pleased with how the manuscript was handled editorially. We look forward to future exciting submissions from your lab.

Sincerely,
